# Ramba: Selective State-Space Models for Relational Deep Learning

**Yiming Liu** [1]  **Chunyu Wei** [1]  **Haozhe Lin** [2]  **Fengjun Xiao** [3]  **Junqi Zhang** [4]  **Yunhai Wang** [1]  **Yueguo Chen** [1]

## Abstract

Relational Deep Learning aims to learn directly on multi-table databases, yet current methods face a fundamental tension: Transformers' quadratic complexity prohibits the large contexts that relational data demands, while GNNs sacrifice global context for efficiency. We introduce Ramba, the first selective state-space model for relational databases. Our approach features two key innovations: (1) Topology-Aware Linearization, which processes cells via global columnar serialization in linear complexity while recovering relational structure through sparse entity and foreign-key attention masks; and (2) Schema Dynamic Gating, which modulates SSM state transitions based on semantic alignment between the currently scanned attribute and the prediction target, enabling cross-table relevance filtering without relying on value distributions. Together, these mechanisms allow Ramba to ingest vast relational contexts while selectively retaining semantically relevant information. Experiments demonstrate state-of-the-art performance with linear scalability across diverse relational benchmarks. [1]

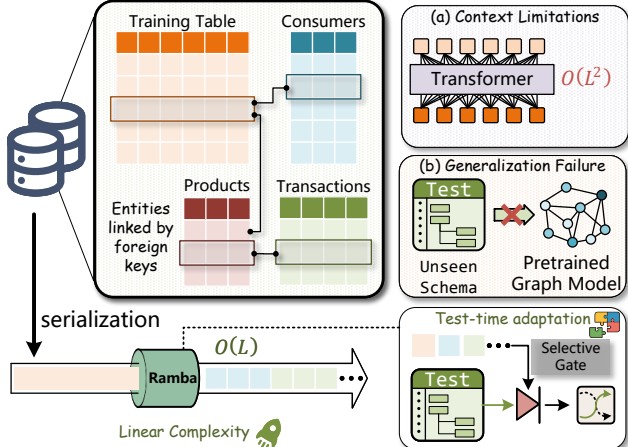

*Figure 1.* **Motivation for RAMBA.** (a) Transformers suffer from $\mathcal{O}(L^2)$ complexity, limiting context capacity. (b) Pretrained graph models fail on unseen schemas. RAMBA addresses both challenges via $\mathcal{O}(L)$ serialization and schema dynamic gating.

## 1. Introduction

Relational databases house majority of the world's structured data, yet machine learning workflows remain bottlenecked by manual feature engineering. Domain experts must hand-craft aggregations and flatten complex multi-table structures into single tables, that is costly, error-prone, and destructive to rich relational semantics. Relational Deep Learning (RDL) aims to eliminate this barrier by training end-to-end models directly on native database structures.

Recent RDL approaches have explored diverse architectures. Graph Neural Networks (GNNs) treat tables as nodes and foreign keys as edges to propagate information, while Transformer-based methods leverage self-attention to model cross-table interactions. Despite this progress, current architectures encounter two fundamental barriers:

**Context Limitation.** Real-world databases contain millions of rows, yet Transformers suffer from quadratic complexity $\mathcal{O}(L^2)$ in sequence length $L$, rendering large-scale processing intractable. GNNs circumvent this cost but sacrifice global context due to limited receptive fields.

**Generalization Failure.** Existing models fail to adapt to unseen schemas at test time, often overfitting to training structures or requiring expensive per-database fine-tuning.

These barriers are deeply intertwined. Robust generalization to novel schemas requires processing extensive contextual information: relational neighborhoods, cross-table dependencies, and distributional patterns across attributes. Quadratic complexity renders this prohibitive. *Efficiency is thus not merely a computational convenience; it is a prerequisite for generalization.*

This insight motivates our central hypothesis: an architec-

[1]School of Information, Renmin University of China, Beijing, China [2]Tsinghua University, Beijing, China [3]Zhejiang Informatization Development Institute, Hangzhou Dianzi University, Hangzhou, China [4]College of Computer Science, Beijing University of Technology, Beijing, China. Correspondence to: Chunyu Wei <weicy15@icloud.com>.

*Proceedings of the $43^{rd}$ International Conference on Machine Learning*, Seoul, South Korea. PMLR 306, 2026. Copyright 2026 by the author(s).

[1]Code is available at https://github.com/ROOOOOOOL/Ramba

ture pairing *linear complexity* with *selective attention* can resolve both barriers simultaneously. Selective state-space models (SSMs), exemplified by Mamba (Gu & Dao, 2023), offer precisely this combination. Operating in $\mathcal{O}(L)$ time, they can ingest the vast contexts that relational databases demand, while their input-dependent gating mechanisms enable selective memory that retains relevant features while filtering noise.

However, adapting SSMs to relational databases presents two fundamental challenges:

- *Serialization Challenge.* Relational data is inherently two-dimensional (rows × columns) with complex foreign-key topology, yet SSMs require one-dimensional input sequences. Naive flattening destroys structural semantics: column-wise distributional patterns become fragmented, and foreign-key relationships are obscured.

- *Adaptation Challenge.* While linear complexity permits ingesting vast contexts, effective prediction requires distinguishing relevant information from noise among thousands of tokens. In relational data, relevance is inherently *schema-dependent*: what constitutes useful context for predicting "price" differs fundamentally from "category," and this relevance must generalize across tables with different value distributions.

We draw inspiration from human cognition in database comprehension. When interpreting relational data, humans naturally combine *sequential reading* (scanning columns to grasp distributional patterns) with *associative jumping* (linking related entities via foreign-key relationships). This cognitive pattern suggests a hybrid architecture that processes global sequences for efficiency while recovering structural dependencies through sparse, targeted interactions.

We introduce RAMBA (**Relational Mamba**), addressing these challenges through two complementary innovations:

**Topology-Aware Linearization.** We propose a two-stage encoding scheme that decouples distributional modeling from structural reasoning. First, *global columnar serialization* concatenates cells column-by-column into a unified sequence with learnable separators, enabling bidirectional SSM dynamics to capture attribute-wise distributional patterns efficiently. Second, *sparse structural attention* recovers relational topology through two complementary masks: an intra-entity mask aggregating attributes within rows, and a relational mask propagating information along foreign-key edges. This decomposition preserves columnar statistics and graph semantics while maintaining $\mathcal{O}(L)$ complexity.

**Schema Dynamic Gating.** We introduce a gating mechanism that modulates SSM selectivity based on semantic alignment between attributes. Rather than relying on value distributions, we compute relevance gates from schema embeddings: the model compares the semantic meaning of the currently scanned column against the target prediction attribute using a frozen language model. This allows the model to determine that a foreign table's "City" column is semantically relevant to predicting "ZipCode" (opening the gate), while "CreationTime" is irrelevant (closing the gate), even across tables with no distributional overlap.

Our contributions are threefold:

(1) We introduce RAMBA, the first selective state-space model for RDL, featuring **Topology-Aware Linearization** that reconciles global sequence modeling with relational structure preservation.

(2) We propose **Schema Dynamic Gating**, enabling cross-table relevance filtering based on attribute semantics for robust generalization to unseen schemas.

(3) We provide empirical validation demonstrating state-of-the-art performance on diverse relational benchmarks while maintaining linear scalability.

**Conflict of Interest Disclosure.** The authors declare no financial conflicts of interest related to this work.

## 2. Related Work

**Relational Deep Learning.** RDL seeks to automate predictive modeling on databases, replacing manual feature engineering with end-to-end learning. Early frameworks (Zhang et al., 2024; Fey et al., 2024) and benchmarks like RelBench (Robinson et al., 2024) established the database-as-graph paradigm as a viable foundation. Subsequent research has expanded this ecosystem through scalable multi-table learning (Yuan et al., 2025), efficient message passing (Chen et al., 2025a), and automated graph construction (Choi et al., 2025; Chen et al., 2025b). While GNNs have dominated these tasks, they often struggle to capture long-range dependencies across multiple foreign-key hops. Conversely, recent Transformer-based approaches (Dwivedi et al., 2025; Meyer et al., 2025; Lachi et al., 2025) offer strong semantic modeling and richer temporal-structural context integration, but incur prohibitive computational costs when processing massive serialized tables.

**Foundation Models for Structured Data.** The success of foundation models in NLP (Brown et al., 2020; Chowdhery et al., 2023) and vision (Dosovitskiy et al., 2021; Radford et al., 2021) has catalyzed efforts to extend these paradigms to structured data. In the tabular domain, recent works (Hollmann et al., 2025; Huang et al., 2020) demonstrate that pre-training enables strong transfer capabilities. This has inspired a nascent wave of *Relational Foundation Models*, including Griffin (Wang et al., 2025), Relational

Transformer (Ranjan et al., 2025), and KumoRFM (Fey et al., 2025), which utilize in-context learning to generalize across diverse schemas. However, these models rely on standard attention mechanisms, inheriting $\mathcal{O}(L^2)$ complexity that fundamentally limits their ability to process large-scale databases without aggressive sampling or truncation.

**Efficient Sequence Modeling.** To address the quadratic bottleneck, substantial research has focused on sub-quadratic architectures. Linear attention approximations—including Performer (Choromanski et al., 2021), Linformer (Wang et al., 2020), and hybrids like RWKV (Peng et al., 2023) and RetNet (Sun et al., 2023)—attempt to scale long-sequence modeling. A parallel line of work utilizes Structured State-Space Models (SSMs), including S4 (Gu et al., 2022), H3 (Fu et al., 2023), and Mamba (Gu & Dao, 2023), achieving $\mathcal{O}(L)$ inference. While earlier sub-quadratic models often lagged behind Transformers in structured reasoning and in-context learning (Arora et al., 2024; Akyürek et al., 2024), Mamba has demonstrated a unique capacity to bridge this gap (Park et al., 2024) via its selective gating mechanism. By adapting bidirectional Mamba to RDL, we combine the generalization potential of foundation models with the linear scalability required for real-world databases.

## 3. Preliminaries

**Relational Database Formalism** A relational database $\mathcal{D} = (\mathcal{T}, \mathcal{F})$ comprises $N$ tables $\mathcal{T} = \{T_1, \ldots, T_N\}$ and foreign-key constraints $\mathcal{F}$. Each table $T_i$ has schema $\mathcal{S}_i = (c_1^{(i)}, \ldots, c_{M_i}^{(i)})$ with $M_i$ columns and data matrix $\mathbf{X}_i \in \mathbb{R}^{n_i \times M_i}$ containing $n_i$ rows. A foreign-key constraint $f = (T_i, c_j, T_{i'}) \in \mathcal{F}$ specifies that column $c_j$ in $T_i$ references the primary key of $T_{i'}$.

The relational structure induces a directed graph $\mathcal{G} = (\mathcal{V}, \mathcal{E})$ where vertices $\mathcal{V}$ represent all rows across tables, and edge $(r, r') \in \mathcal{E}$ indicates that row $r$ contains a foreign key referencing $r'$. We define a *cell* as an atomic data unit $\xi = (v, c, \tau, T)$ comprising value $v$, column $c$, data type $\tau \in \{\texttt{num}, \texttt{cat}, \texttt{temp}\}$, and source table $T$. For row $r$, we denote its constituent cells as $\Xi(r)$.

**Selective State-Space Models** State-space models map input sequences to outputs through latent dynamics. The continuous system $\frac{d\mathbf{h}}{dt} = \mathbf{A}\mathbf{h} + \mathbf{B}x$, $y = \mathbf{C}\mathbf{h}$ is discretized via zero-order hold with step size $\Delta$:

$$\bar{\mathbf{A}} = \exp(\Delta\mathbf{A}), \quad \bar{\mathbf{B}} = (\Delta\mathbf{A})^{-1}(\exp(\Delta\mathbf{A}) - \mathbf{I}) \cdot \Delta\mathbf{B} \quad (1)$$

yielding the recurrence $\mathbf{h}_t = \bar{\mathbf{A}}\mathbf{h}_{t-1} + \bar{\mathbf{B}}x_t$ and output $y_t = \mathbf{C}\mathbf{h}_t$. Selective SSMs (Gu & Dao, 2023) make system matrices input-dependent. For input $\mathbf{x}_t \in \mathbb{R}^D$:

$$\Delta_t = \texttt{softplus}(\mathbf{W}_\Delta\mathbf{x}_t + \mathbf{b}_\Delta), \mathbf{B}_t = \mathbf{W}_B\mathbf{x}_t, \mathbf{C}_t = \mathbf{W}_C\mathbf{x}_t \tag{2}$$

where $\mathbf{W}_\Delta, \mathbf{W}_B, \mathbf{W}_C \in \mathbb{R}^{D_h \times D}$ are learnable projections and $D_h$ is the state dimension. The discretized matrices become $\bar{\mathbf{A}}_t = \exp(\text{diag}(\Delta_t)\mathbf{A})$ and $\bar{\mathbf{B}}_t = \text{diag}(\Delta_t)\mathbf{B}_t$, with diagonal state matrix $\mathbf{A} \in \mathbb{R}^{D_h}$.

The selectivity mechanism is crucial: when $\Delta_t \to \mathbf{0}$, we have $\bar{\mathbf{A}}_t \to \mathbf{I}$ and $\bar{\mathbf{B}}_t \to \mathbf{0}$, causing the state to pass unchanged ($\mathbf{h}_t \approx \mathbf{h}_{t-1}$). This enables content-dependent filtering exploited for schema-based context selection.

## 4. Methodology

We propose RAMBA, inspired by human cognitive processes for database comprehension. When interpreting relational data, humans naturally combine *sequential reading* (scanning columns to grasp distributional patterns) with *associative jumping* (linking related entities via foreign-key relationships). RAMBA operationalizes this intuition through three stages: Global Columnar Serialization (§4.2), Schema Dynamic Gating (§4.3), and Sparse Structural Attention (§4.4). Figure 2 provides an architectural overview.

### 4.1. Subgraph Sampling and Cell Embedding

Given target row $r^* \in \mathcal{V}$, we construct a context subgraph $\mathcal{G}_{\text{sub}} \subseteq \mathcal{G}$ via breadth-first traversal to depth $K$. This enforces three constraints: (i) *full parent inclusion* for foreign-key targets, (ii) *time-restricted child sampling* within lookback window $[t(r^*) - \Delta\tau, t(r^*)]$ with budget $N_{\text{child}}$, and (iii) *strict causality* excluding rows where $t(r') > t(r^*)$.

Each cell $\xi = (v, c, \tau, T)$ is embedded by fusing schema and value information. The schema embedding leverages a frozen language model: $\mathbf{e}_{\text{schema}} = \phi_{\text{LM}}(\texttt{concat}(T, \texttt{": "}, c))$. Value embeddings are type-specific: Piecewise Linear Encoding for numerics, learned embeddings for categoricals, and sinusoidal encoding for timestamps. The unified cell embedding is:

$$\mathbf{x}_\xi = \mathbf{W}_v\mathbf{e}_{\text{value}} + \mathbf{W}_s\mathbf{e}_{\text{schema}} + \mathbf{W}_c\mathbf{e}_{\text{scale}} \in \mathbb{R}^D \quad (3)$$

where $\mathbf{W}_v \in \mathbb{R}^{D \times D_v}$ and $\mathbf{W}_s \in \mathbb{R}^{D \times D_{\text{LM}}}$ and $\mathbf{W}_c \in \mathbb{R}^{D \times D_{\text{scale}}}$ are projections.

### 4.2. Global Columnar Serialization

To map the two-dimensional relational structure into a one-dimensional stream processable by SSMs, we adopt a **global serialization** strategy rather than processing columns independently. We concatenate all cells from the subgraph column-by-column, forming a single sequence containing the complete context. Let the subgraph involve $K$ attribute columns $\{c_1, c_2, \ldots, c_K\}$. For column $c_k$, let

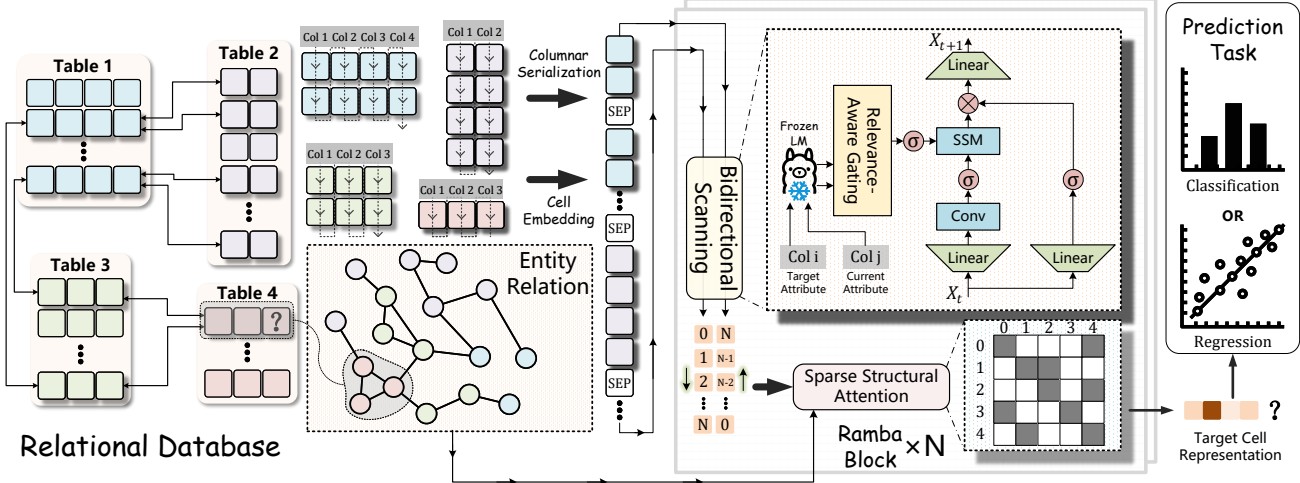

*Figure 2.* **RAMBA Architecture.** A relational database is serialized column-by-column with [SEP] tokens, then processed by stacked RAMBA blocks. Each block applies bidirectional scanning with Schema Dynamic Gating followed by Sparse Structural Attention (intra-entity and relational masks). The target cell representation is decoded for classification or regression.

$\mathcal{C}_k = \{\xi : c(\xi) = c_k, \xi \in \Xi_{\text{sub}}\}$ denote its cell set. The global input sequence is:

$$\mathcal{S} = [\text{SEP}_1, \mathcal{C}_1, \text{SEP}_2, \mathcal{C}_2, \dots, \text{SEP}_K, \mathcal{C}_K] \quad (4)$$

where $\text{SEP}_k \in \mathbb{R}^D$ are learnable separator tokens signaling column transitions. This arrangement enables the model to traverse all entity attributes within a unified sequence, establishing global context for subsequent processing. The total sequence length is $L = K + |\Xi_{\text{sub}}|$.

### 4.3. Schema Dynamic Gating

Standard SSMs process sequences uniformly, which is suboptimal for relational data where noise from irrelevant columns in distant tables can overwhelm useful signal. We introduce **Schema Dynamic Gating**, which modulates state transitions based on semantic alignment between the currently scanned attribute and the target prediction attribute.

**Semantic Context Definition.** Let the prediction target be a cell in attribute $c^*$. We define the *target semantic context* $\mathbf{s}_{\text{tgt}}$ using a frozen language model $\phi_{\text{LM}}$ on the target attribute's metadata $\mathbf{s}_{\text{tgt}} = \phi_{\text{LM}}(\text{desc}(c^*))$, where $\text{desc}(c^*)$ denotes the column name and table description of $c^*$. As the model scans the global sequence $\mathcal{S}$, the *current semantic context* $\mathbf{s}_{\text{curr}}(t)$ updates dynamically. When the scan crosses separator $\text{SEP}_k$ and enters the region of column $c_k$, the context shifts: $\quad \mathbf{s}_{\text{curr}}(t) = \phi_{\text{LM}}(\text{desc}(c_k)), \quad \text{if } x_t \in \mathcal{C}_k.$

This ensures that whether scanning a local "Price" column or a foreign "Supplier_Region" column, the model remains aware of the semantic domain it is traversing.

**Relevance-Aware Gating.** We compute a semantic relevance gate $g_t^{\text{sem}}$ measuring the utility of the current attribute

$c_k$ for predicting target $c^*$:

$$g_t^{\text{sem}} = \sigma\left(\mathbf{w}_g^\top \cdot \text{GELU}(\mathbf{W}_{\text{rel}}[\mathbf{s}_{\text{curr}}(t); \mathbf{s}_{\text{tgt}}]) + b_g\right) \in (0, 1)$$

where $\mathbf{W}_{\text{rel}} \in \mathbb{R}^{D_g \times 2D_{\text{LM}}}$ projects the concatenated semantic contexts, $\mathbf{w}_g \in \mathbb{R}^{D_g}$, and $\sigma$ is the sigmoid function.

Crucially, this gate relies solely on schema semantics, not cell values. This enables the model to determine that a foreign table's "City" column is semantically relevant to a target "ZipCode" (opening the gate), while "CreationTime" might be irrelevant (closing the gate), even when specific values have not been encountered during training.

**Modulated State Dynamics.** We integrate the semantic gate into the SSM discretization process (Eq. 2). The discretization step $\Delta_t$ is modulated by our semantic gate:

$$\tilde{\Delta}_t = \text{softplus}(\mathbf{W}_\Delta \mathbf{x}_t + \mathbf{b}_\Delta) \odot g_t^{\text{sem}} \quad (5)$$

The discretized system matrices become:

$$\tilde{\bar{\mathbf{A}}}_t = \exp(\text{diag}(\tilde{\Delta}_t)\mathbf{A}), \quad \tilde{\bar{\mathbf{B}}}_t = \text{diag}(\tilde{\Delta}_t)\mathbf{B}_t \quad (6)$$

By scaling $\Delta_t$ with $g_t^{\text{sem}}$, we control the rate at which the hidden state evolves. When $g_t^{\text{sem}} \to 0$ (semantically irrelevant column), $\tilde{\Delta}_t \to \mathbf{0}$, causing $\tilde{\bar{\mathbf{A}}}_t \to \mathbf{I}$ and $\tilde{\bar{\mathbf{B}}}_t \to \mathbf{0}$. This effectively "freezes" the hidden state, preserving memory of relevant past information while bypassing noise from distant foreign tables.

**Bidirectional Scanning.** Unidirectional SSMs suffer from causal bias: tokens late in the sequence cannot influence representations of earlier tokens. For relational data lacking inherent ordering, this asymmetry is undesirable. We apply bidirectional scanning, processing $\mathcal{S}$ in both forward and

backward directions:

$$\overrightarrow{\mathbf{H}} = \text{GatedSSM}^{\rightarrow}(\mathcal{S}), \quad \overleftarrow{\mathbf{H}} = \text{GatedSSM}^{\leftarrow}(\mathcal{S}) \quad (7)$$

where $\text{GatedSSM}^{\leftarrow}$ processes the reversed sequence. The bidirectional outputs are merged via learned projection:

$$\mathbf{H} = \mathbf{W}_{\text{merge}}[\overrightarrow{\mathbf{H}}; \overleftarrow{\mathbf{H}}] + \mathbf{b}_{\text{merge}} \in \mathbb{R}^{L \times D} \quad (8)$$

where $\mathbf{W}_{\text{merge}} \in \mathbb{R}^{D \times 2D_h}$ and $[\cdot; \cdot]$ denotes concatenation along the feature dimension. This ensures each cell's representation incorporates context from the entire sequence regardless of position.

## 4.4. Sparse Structural Attention

Linear scanning does not explicitly encode the graph's topological structure (foreign-key references, entity boundaries). To recover these structural dependencies, we introduce **Sparse Structural Attention** with two complementary masks that maintain $\mathcal{O}(L)$ complexity.

**Intra-Entity Mask.** $\mathbf{M}_{\text{intra}} \in \{0, 1\}^{L \times L}$ allows interaction only among cells belonging to same rows:

$$[\mathbf{M}_{\text{intra}}]_{ij} = \mathbb{1}[\xi_i, \xi_j \in \Xi(r) \text{ for some } r \in \mathcal{V}_{\text{sub}}] \quad (9)$$

ensuring attributes of the same entity (e.g., "user_id" and "name" from one row) can aggregate, preserving integrity.

**Relational Mask.** $\mathbf{M}_{\text{rel}} \in \{0, 1\}^{L \times L}$ allows interaction only between cells connected by foreign-key relationships:

$$[\mathbf{M}_{\text{rel}}]_{ij} = \mathbb{1}[\xi_i \in \Xi(r), \xi_j \in \Xi(r'), (r, r') \in \mathcal{E}_{\text{sub}}] \quad (10)$$

This enables information flow along relational edges, corresponding to FK $\rightarrow$ PK references in the database schema.

For input $\mathbf{H}$ from the Mamba layer, sparse attention with mask $\mathbf{M} \in \{\mathbf{M}_{\text{intra}}, \mathbf{M}_{\text{rel}}\}$ computes:

$$\begin{aligned}&\text{SparseAttn}(\mathbf{H}, \mathbf{M}) \\ =&\text{softmax}\left(\frac{\mathbf{Q}\mathbf{K}^{\top}}{\sqrt{D_k}} \odot \mathbf{M} + (\mathbf{1} - \mathbf{M}) \cdot (-\infty)\right) \mathbf{V}\end{aligned} \quad (11)$$

where $\mathbf{Q} = \mathbf{H}\mathbf{W}_Q$, $\mathbf{K} = \mathbf{H}\mathbf{W}_K$, $\mathbf{V} = \mathbf{H}\mathbf{W}_V$. Since the number of neighbors per cell is bounded by the graph sampling degree (constant with respect to $L$), computational cost remains linear.

## 4.5. RAMBA Block and Overall Architecture

A complete RAMBA block integrates the above modules with residual connections and layer normalization:

$$\begin{aligned}\mathbf{H}' &= \mathbf{X} + \text{SchemaDynamicGatedMamba}(\text{LN}(\mathbf{X})) \\ \mathbf{H}'' &= \mathbf{H}' + \text{SparseAttn}(\text{LN}(\mathbf{H}'), \mathbf{M}_{\text{intra}}) \\ \mathbf{H}_{\text{out}} &= \mathbf{H}'' + \text{SparseAttn}(\text{LN}(\mathbf{H}''), \mathbf{M}_{\text{rel}})\end{aligned}$$

We stack $L_{\text{layers}}$ such blocks. Each layer absorbs global distributional information through Mamba and performs precise topological reasoning through sparse attention, operationalizing the "reading + association" cognitive pattern.

**Complexity Analysis.** Let $L$ denote sequence length, $D$ the hidden dimension, and $d_{\max}$ the maximum number of columns in $\mathcal{G}_{\text{sub}}$. The Schema Dynamic Gated Mamba incurs $\mathcal{O}(L \cdot D \cdot D_h)$ cost (linear in $L$). Each sparse attention layer costs $\mathcal{O}(L \cdot d_{\max} \cdot D)$, which is $\mathcal{O}(L)$ when $d_{\max}$ is constant. The total per-block complexity is $\mathcal{O}(L \cdot D \cdot \max(D_h, d_{\max}))$, enabling processing of contexts orders of magnitude larger than quadratic alternatives.

**Permutation Invariance.** RAMBA maintains invariance to row permutations within tables. Global serialization orders cells by column (not row), and sparse attention masks depend only on graph structure $\mathcal{E}_{\text{sub}}$, not vertex indexing.

## 4.6. Prediction and Training

After $L_{\text{layers}}$ RAMBA blocks, we obtain final representations $\mathbf{Z} = \mathbf{H}_{\text{out}}^{(L_{\text{layers}})} \in \mathbb{R}^{L \times D}$. For target cell $\xi^*$ with embedding $\mathbf{z}^* \in \mathbb{R}^D$, we apply type-specific heads.

For numerical targets:

$$\hat{y}^* = (\mathbf{w}_{\text{reg}}^{\top}\text{MLP}(\mathbf{z}^*) + b_{\text{reg}}) \cdot \sigma_{c^*} + \mu_{c^*} \quad (12)$$

where $\mu_{c^*}$ and $\sigma_{c^*}$ are column statistics for scale invariance. For categorical targets with vocabulary $\mathcal{V}_{c^*}$:

$$\hat{\mathbf{p}}^* = \text{softmax}(\mathbf{W}_{\text{cls}}\text{MLP}(\mathbf{z}^*) + \mathbf{b}_{\text{cls}}) \in \mathbb{R}^{|\mathcal{V}_{c^*}|} \quad (13)$$

We train with a unified objective over masked cells $\mathcal{M}$:

$$\mathcal{L} = \frac{1}{|\mathcal{M}|} \sum_{\xi \in \mathcal{M}} [\mathbb{1}_{\tau(\xi)=\text{num}}\mathcal{L}_{\text{Huber}}(y_\xi, \hat{y}_\xi) + \mathbb{1}_{\tau(\xi)=\text{cat}}\mathcal{L}_{\text{CE}}(y_\xi, \hat{\mathbf{p}}_\xi)]$$

where Huber loss provides robustness to outliers and cross-entropy handles categorical distributions.

## 5. Theoretical Analysis

By modulating the SSM discretization step $\Delta_t$ with schema-based relevance gates, RAMBA can selectively preserve information from semantically aligned columns while suppressing noise from irrelevant attributes, which is essential for generalization across heterogeneous relational schemas.

Consider a relational prediction task where the target attribute $c^*$ depends on a subset of semantically related columns $\mathcal{R} \subseteq \{c_1, \ldots, c_K\}$ across potentially multiple tables. We model semantic relevance through an alignment function $\rho : \mathcal{C} \times \mathcal{C} \rightarrow [0, 1]$ derived from schema embeddings, where $\rho(c_k, c^*) \approx 1$ if column $c_k$ is semantically relevant to predicting $c^*$, and $\rho(c_k, c^*) \approx 0$ otherwise.

**Theorem 5.1** (Schema-Guided Context Selection). *Let $\mathcal{S} = [SEP_1, \mathcal{C}_1, \ldots, SEP_K, \mathcal{C}_K]$ be a serialized relational context of length $L$ with $K$ attribute columns, where column $c_k$ contains $n_k$ cells with embeddings $\{\mathbf{x}_{k,i}\}_{i=1}^{n_k}$. Let $\mathcal{R} \subseteq [K]$ denote indices of semantically relevant columns with $|\mathcal{R}| = r \ll K$. Suppose the Schema Dynamic Gating mechanism (Eq. 4.3) produces gates $g_t^{\text{sem}}$ satisfying:*

*(i) **Relevance detection:** For cells in relevant columns ($t \in \mathcal{C}_k, k \in \mathcal{R}$): $g_t^{\text{sem}} \geq 1 - \epsilon$ for some $\epsilon \in (0, 1/2)$.*

*(ii) **Noise suppression:** For cells in irrelevant columns ($t \in \mathcal{C}_k, k \notin \mathcal{R}$): $g_t^{\text{sem}} \leq \delta$ for some $\delta \in (0, \epsilon)$.*

*Then the output $\mathbf{h}_L$ of the gated SSM satisfies:*

$$\mathbf{h}_L = \sum_{k \in \mathcal{R}} \sum_{i=1}^{n_k} \alpha_{k,i} \mathbf{B}_{k,i} x_{k,i} + \mathbf{E}_{\text{noise}}, \quad (14)$$

*where the attention weights $\alpha_{k,i} = \Omega(1/n_{\mathcal{R}})$ with $n_{\mathcal{R}} = \sum_{k \in \mathcal{R}} n_k$, and the noise term is bounded as $\|\mathbf{E}_{\text{noise}}\| \leq O(\delta L \cdot \max_t \|\mathbf{B}_t x_t\|)$. Consequently, the signal-to-noise ratio scales as $\text{SNR} = \Omega\left(\frac{(1-\epsilon)n_{\mathcal{R}}}{\delta L}\right)$.*

*Proof.* We analyze the gated SSM dynamics with modulated discretization $\tilde{\Delta}_t = \Delta_t \odot g_t^{\text{sem}}$ (Eq. 5). For diagonal state matrix $\mathbf{A} = -\mathbf{I}_{D_h}$, the discretized transition becomes $\tilde{\bar{\mathbf{A}}}_t = \exp(-\tilde{\Delta}_t)$ and $\tilde{\bar{\mathbf{B}}}_t = (1 - \exp(-\tilde{\Delta}_t)) \odot \mathbf{B}_t$. The hidden state evolves as $\mathbf{h}_t = \tilde{\bar{\mathbf{A}}}_t \mathbf{h}_{t-1} + \tilde{\bar{\mathbf{B}}}_t x_t$. Unrolling this recurrence:

$$\mathbf{h}_L = \sum_{t=1}^{L} \left( \prod_{s=t+1}^{L} \tilde{\bar{\mathbf{A}}}_s \right) \tilde{\bar{\mathbf{B}}}_t x_t = \sum_{t=1}^{L} G_{t,L} \tilde{\bar{\mathbf{B}}}_t x_t, \quad (15)$$

where $\quad G_{t,L} = \prod_{s=t+1}^{L} \exp(-\tilde{\Delta}_s) = \exp\left(-\sum_{s=t+1}^{L} \tilde{\Delta}_s\right).$

**Case 1: Relevant columns ($k \in \mathcal{R}$).** For $t \in \mathcal{C}_k$, we have $g_t^{\text{sem}} \geq 1 - \epsilon$, so $\tilde{\Delta}_t \geq (1-\epsilon)\Delta_t$. The input contribution satisfies $\|\tilde{\bar{\mathbf{B}}}_t x_t\| \geq (1 - e^{-(1-\epsilon)\Delta_t})\|\mathbf{B}_t x_t\| = \Omega(\Delta_t \|\mathbf{B}_t x_t\|)$ for bounded $\Delta_t$. For the last relevant cell at position $t^* = \max\{t : t \in \mathcal{C}_k, k \in \mathcal{R}\}$, we have $G_{t^*,L} \geq \exp(-\delta(L - t^*)\bar{\Delta}) = \Omega(1)$ when $\delta L \bar{\Delta} = O(1)$.

**Case 2: Irrelevant columns ($k \notin \mathcal{R}$).** For $t \in \mathcal{C}_k$, we have $g_t^{\text{sem}} \leq \delta$, yielding $\tilde{\Delta}_t \leq \delta \Delta_t$. The input contribution is suppressed: $\|\tilde{\bar{\mathbf{B}}}_t x_t\| \leq (1 - e^{-\delta \Delta_t})\|\mathbf{B}_t x_t\| \leq \delta \Delta_t \|\mathbf{B}_t x_t\| = O(\delta \|\mathbf{B}_t x_t\|)$.

Decomposing $\mathbf{h}_L$ into relevant and irrelevant contributions yields the stated bounds. The SNR follows from $\text{SNR} = \frac{\|\text{signal}\|}{\|\mathbf{E}_{\text{noise}}\|} \geq \frac{\Omega(n_{\mathcal{R}}(1-\epsilon)\bar{B})}{O(\delta L \bar{B})} = \Omega\left(\frac{(1-\epsilon)n_{\mathcal{R}}}{\delta L}\right).$ $\quad\square$

When the gating network successfully distinguishes relevant from irrelevant columns, the SNR scales favorably with the fraction of relevant data $n_{\mathcal{R}}/L$, independent of total context size. Since gates derive from schema semantics rather than value distributions, this filtering generalizes to unseen schemas where column names and table descriptions provide sufficient semantic signal.

# 6. Experiments

**Datasets and Tasks.** We evaluate on RelBench (Robinson et al., 2024), comprising seven real-world relational databases: *rel-amazon*, *rel-hm*, *rel-stack*, *rel-avito*, *rel-f1*, *rel-trial*, and *rel-event*. Each database defines multiple forecasting tasks spanning binary classification and regression. Following prior work (Ranjan et al., 2025), we omit *rel-event* due to documented temporal leakage issues.

**Evaluation Protocol.** We adopt the *leave-one-database-out* protocol: for each evaluation, we pretrain on six databases and test on the held-out one, rotating through all seven. We report AUROC for classification and $R^2$ for regression.

**Baselines.** We compare against: *Schema-agnostic* methods (cross-database transferable): **Griffin** (Wang et al., 2025), **Relational Transformer (RT)** (Ranjan et al., 2025), and **RelLLM** (Wu et al., 2025); and *Schema-specific* methods (task-specific training): **RDL-GNN** (Fey et al., 2024), **Rel-GNN** (Chen et al., 2025a), **RelGT** (Dwivedi et al., 2025), **LightGBM**, and **EntityMean**.

**Implementation.** We pretrain RAMBA for 50k steps with batch size 64 using AdamW (weight decay 0.1) and linear schedule with 20% warmup (peak lr: $5 \times 10^{-4}$). Fine-tuning uses 33k steps with lr $10^{-4}$. All experiments use $2 \times$A800 GPUs with BFloat16 precision. Unless otherwise stated, all pretrainable baselines are reproduced under the same RelBench splits, context length, sampling budget, hardware, and BFloat16 precision. In particular, the main comparisons use the same subgraph sampling protocol across Griffin, RT, and RAMBA; additional sampling variants are reported in Appendix B.3.7.

## 6.1. Zero-Shot Cross-Database Generalization

**Zero-shot setting.** In our setting, **zero-shot** denotes *cross-database generalization*: after pretraining on source databases, the model is evaluated on a held-out database with unseen tables, columns, foreign-key structure, and value distributions, without any weight updates on that target database. The task definition still specifies the output format, such as the number of classes or the regression target. The challenge is therefore to transfer relational reasoning to an unseen schema, rather than to infer an unknown label space.

Tables 1 and 2 present zero-shot performance alongside fine-tuned results.

*Table 1.* **Zero-shot and fine-tuned classification results (AUROC %). Bold**: best in category; underline: best overall.

| Dataset | Task | LLM Baselines | | | Zero-Shot (Pretrained) | | | | Schema-Specific | | | | | Fine-Tuned | | |
|---|---|---|---|---|---|---|---|---|---|---|---|---|---|---|---|---|
| | | Gemma-4B | Gemma-12B | Gemma-27B | RelLLM | Griffin | RT | **Ramba** | EntMean | LGBM | RDL-GNN | RelGNN | RelGT | Griffin | RT | **Ramba** |
| *Parameters* | | 4B | 12B | 27B | 3B | 22M | 22M | **12M** | – | – | 13M | 88M | 17M | 22M | 22M | **12M** |
| rel-amazon | item-churn | 62.1 | 55.0 | 42.1 | 64.1 | 69.0 | 70.2 | **71.1** | 73.0 | 57.2 | 79.2 | 56.4 | 77.3 | **79.9** | 77.9 | 78.0 |
| rel-amazon | user-churn | 58.1 | 54.7 | 50.5 | 60.1 | 62.3 | 63.9 | **64.9** | 64.4 | 51.8 | 64.4 | 57.2 | 62.6 | **69.4** | 67.2 | 67.5 |
| rel-avito | user-clicks | 54.5 | 59.5 | 59.8 | **62.3** | 45.9 | 58.0 | 59.5 | 44.7 | 50.8 | 62.2 | 59.8 | 59.2 | **64.7** | 55.9 | 60.5 |
| rel-avito | user-visits | 60.1 | 57.9 | 62.7 | 56.2 | 60.7 | **61.4** | 61.2 | 60.7 | 50.0 | **64.9** | 61.6 | 60.1 | 62.6 | 59.4 | 60.9 |
| rel-f1 | driver-dnf | 56.2 | 54.6 | 75.8 | 71.8 | 57.7 | 77.2 | **80.5** | 75.4 | 67.6 | 71.8 | 67.3 | 70.9 | 66.7 | 82.0 | **82.1** |
| rel-f1 | driver-top3 | 84.6 | 90.5 | 91.4 | 70.6 | 82.5 | **89.1** | 87.4 | 85.0 | 66.1 | 65.5 | 82.7 | 75.8 | 78.7 | 89.4 | **91.3** |
| rel-hm | user-churn | 59.8 | 47.1 | 48.7 | 56.0 | 60.2 | 62.8 | **69.4** | 64.4 | 52.1 | 68.7 | 59.2 | 62.7 | 68.0 | **70.5** | **70.5** |
| rel-stack | user-badge | 79.1 | 79.8 | 80.0 | 62.1 | 73.5 | **79.0** | 78.0 | 66.2 | 56.6 | 85.0 | 81.1 | 80.0 | **87.0** | 78.7 | 80.7 |
| rel-stack | user-engage | 65.9 | 67.8 | 78.0 | 69.5 | 77.5 | 77.1 | **91.3** | 83.5 | 61.2 | 86.3 | 80.8 | 80.4 | 90.4 | 92.3 | **93.9** |
| rel-trial | study-out | 52.6 | 57.4 | 57.2 | 59.0 | 51.0 | 54.5 | **61.6** | 50.0 | 65.0 | 62.0 | 60.4 | 60.9 | 64.6 | **70.6** | 67.5 |
| **Average** | | 63.3 | 62.4 | 64.6 | 63.2 | 64.0 | 69.3 | **72.5** | 66.7 | 57.8 | 71.0 | 66.6 | 69.0 | 73.2 | 74.4 | **75.3** |

*Table 2.* **Zero-shot and fine-tuned regression results ($R^2$ %).** Higher is better; global mean baseline yields $R^2 = 0$. **Bold**: best in category; underline: best overall.

| Dataset | Task | Zero-Shot (Pretrained) | | | Schema-Specific | | | | | Fine-Tuned | | |
|---|---|---|---|---|---|---|---|---|---|---|---|---|
| | | Griffin | RT | **Ramba** | EntMean | LGBM | RDL-GNN | RelGNN | RelGT | Griffin | RT | **Ramba** |
| rel-amazon | item-ltv | 20.1 | 33.2 | **54.3** | 54.2 | −9.2 | 0.4 | −1.1 | 2.0 | 25.2 | 17.0 | **69.5** |
| rel-amazon | user-ltv | 20.6 | **33.9** | 33.1 | 19.9 | −0.5 | 12.7 | −9.2 | 6.9 | 32.9 | 28.0 | **36.6** |
| rel-avito | ad-ctr | 2.4 | 4.5 | **9.3** | 3.4 | −5.9 | 8.98 | 2.67 | 11.7 | 8.4 | **11.8** | 10.5 |
| rel-f1 | driver-pos | −0.7 | **54.7** | 47.8 | 38.2 | 0.33 | 11.8 | 0.06 | −4.4 | 0.6 | **55.9** | 54.1 |
| rel-hm | item-sales | 2.7 | 14.0 | **14.1** | 1.8 | −1.7 | 11.5 | −1.7 | 17.8 | 30.4 | 35.6 | **41.2** |
| rel-stack | post-votes | 27.4 | 32.2 | **36.3** | 43.7 | −3.4 | 2.34 | −3.4 | −3.4 | 42.7 | 35.2 | **40.6** |
| rel-trial | site-succ | 1.4 | 2.7 | **5.7** | −6.4 | −15.7 | −42.9 | −8.8 | −40.2 | −2.4 | −3.3 | **8.0** |
| rel-trial | study-adv | −2.5 | −0.1 | **1.3** | −0.5 | 16.5 | 10.4 | 0.1 | 12.7 | 18.2 | 34.1 | **45.7** |
| **Average** | | 8.9 | 21.9 | **25.2** | 19.3 | −2.4 | 1.9 | −2.7 | 0.4 | 19.5 | 26.8 | **38.3** |

*Table 3.* **Module ablation.** We ablate three components: Mamba backbone, intra-entity attention (`int`), and relational attention (`rel`). $\Delta$ measures the gain from enabling Mamba.

| | | Classification (AUROC %) | | | Regression ($R^2$ %) | | |
|---|---|---|---|---|---|---|---|
| int | rel | off | on | $\Delta$ | off | on | $\Delta$ |
| ✗ | ✗ | 50.0 | 68.5 | +18.5 | 0.0 | 8.2 | +8.2 |
| ✗ | ✓ | 50.0 | 71.2 | +21.2 | 0.0 | 13.4 | +13.4 |
| ✓ | ✗ | 55.0 | 71.6 | +16.6 | −0.4 | 17.0 | +17.4 |
| ✓ | ✓ | 72.8 | **72.8** | +0.0 | 17.5 | **25.2** | +7.7 |

**Classification Results.** Table 1 shows that RAMBA achieves the highest average zero-shot AUROC (72.5%) among all pretrainable methods, outperforming RT (69.3%) and Griffin (64.0%) by substantial margins. After fine-tuning, RAMBA further extends its lead. Notably, RAMBA excels on tasks requiring cross-table reasoning: on *rel-hm/user-churn* (+6.6% over RT) and *rel-stack/user-engage* (+14.2% over RT), where Schema Dynamic Gating effectively filters semantically relevant columns from foreign tables.

**Regression Results.** Table 2 demonstrates even more pronounced advantages for regression. In zero-shot evaluation, RAMBA achieves 25.2% average $R^2$ in zero-shot evaluation, compared to 21.9% for RT and 8.9% for Griffin. The improvement is particularly striking on *rel-amazon/item-ltv* (54.3% vs. 33.2%), where accurate lifetime value prediction requires integrating numerical signals across multiple related tables. After fine-tuning, the gap widens further: RAMBA attains 38.3% average $R^2$, substantially outperforming RT (26.8%) and Griffin (19.5%)—a **43% relative improvement** over the strongest baseline.

**Comparison with LLM-based Methods.** Despite having 3B parameters (vs. 12M for RAMBA), RelLLM achieves only 63.2% average AUROC, highlighting that raw language modeling capacity does not directly translate to relational reasoning ability. RAMBA's architectural inductive biases prove more effective than scale alone.

To assess robustness across random seeds, we repeat representative pretraining runs with five seeds and compare RAMBA against the strongest zero-shot baseline, RT. RAMBA's improvements are statistically significant on most key tasks, including rel-amazon/item-churn, rel-hm/user-churn, rel-stack/user-engage, rel-amazon/item-ltv, rel-stack/post-votes, and rel-trial/site-succ. Full mean, standard deviation, and $p$-values are reported in Appendix B.1.

### 6.2. Extension to RelBench v2: SALT

To evaluate whether RAMBA generalizes beyond the original RelBench suite, we conduct additional experiments on SALT (Klein et al., 2025), a RelBench v2 ERP database with eight multi-class classification tasks. Unlike many public benchmarks with relatively descriptive schema names, SALT contains domain-specific business attributes such as `SALESOFFICE` and `HEADERINCOTERMSCLASSIFICATION`. This makes it a useful stress test for whether Schema Dynamic Gating can operate under technical industrial naming conventions rather than relying only on clean natural-language column names.

Table 4 reports Mean Reciprocal Rank (MRR) on all eight SALT tasks. We compare RAMBA with GraphSAGE, HGT, and RT under the same evaluation protocol. RAMBA achieves the best average MRR of 0.77, outperforming GraphSAGE by 0.10, HGT by 0.08, and RT by 0.02. The gains are most visible on non-saturated tasks such as *item-incoterms*, *sales-group*, *sales-payterms*, and *sales-shipcond*,

*Table 4.* MRR Results on SALT from RelBench v2 for eight ERP classification tasks. **Bold** indicates the best result for each task.

| Task | GraphSAGE | HGT | RT | RAMBA |
|---|---|---|---|---|
| item-incoterms | 0.64 | 0.74 | 0.77 | **0.81** |
| item-plant | **0.99** | **0.99** | **0.99** | **0.99** |
| item-shippoint | 0.97 | **0.99** | **0.99** | **0.99** |
| sales-group | 0.20 | 0.12 | 0.23 | **0.26** |
| sales-incoterms | 0.59 | 0.66 | **0.76** | **0.76** |
| sales-office | **0.99** | **0.99** | **0.99** | **0.99** |
| sales-payterms | 0.39 | 0.39 | 0.56 | **0.57** |
| sales-shipcond | 0.59 | 0.63 | 0.73 | **0.75** |
| Average | 0.67 | 0.69 | 0.75 | **0.77** |

where cross-table reasoning and schema-aware filtering are more important. These results suggest that RAMBA's advantage is not restricted to the original RelBench databases and that Schema Dynamic Gating remains effective under industrial ERP-style schema naming.

### 6.3. Ablation Studies

**Module Ablation** Table 3 examines the interplay between components. Key findings: (1) **Mamba provides global context integration**—with sparse attention disabled, enabling Mamba yields +18.5% AUROC and +8.2% $R^2$; (2) **Sparse attention captures structural semantics**—enabling both masks without Mamba achieves 72.8% AUROC; (3) **Synergistic combination for regression**—the full model achieves 25.2% $R^2$, a +7.7% gain over attention-only, confirming that fine-grained numerical prediction requires both global distributional modeling and structural reasoning.

**Serialization Strategy** Figure 3 compares serialization strategies. Column-wise serialization performs best. Notably, column-wise with within-column shuffling retains 95% performance, confirming that Mamba relies on column-level grouping rather than strict within-column ordering. Row-wise and random orderings degrade more substantially, indicating that the "column-first" inductive bias is important for capturing attribute-wise distributional patterns.Additional column-order perturbation experiments further show that RAMBA does not rely on brittle schema-order shortcuts: randomizing column blocks or shuffling values within columns causes only minor degradation. Detailed results are provided in Appendix B.3.3.

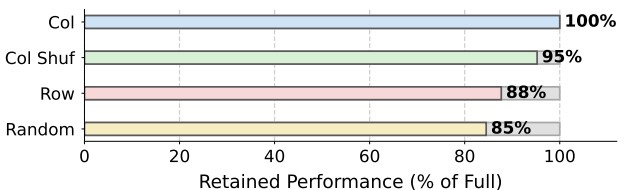

*Figure 3.* **Serialization ablation.**

**Schema Dynamic Gating** Figure 4a ablates gating variants. Schema-based gating achieves the best trade-off, with particularly large gains on regression (+2.7% $R^2$ over no gating). Crucially, shuffled column names perform *worse* than no gating, demonstrating that incorrect semantic alignment is more harmful than no alignment—the mechanism actively uses schema information rather than learning spurious correlations.

We further stress-test schema quality by replacing column names with generic, random, or empty identifiers. RAMBA degrades gracefully rather than collapsing: even when column names are removed, the zero-shot AUROC only drops from 72.5 to 70.9, indicating that schema semantics are beneficial but not the sole source of performance. Full results are reported in Appendix B.3.5.

**Subgraph Sampling** Figure 4b compares sampling strategies. Width-limited BFS with foreign-key prioritization consistently outperforms full BFS, equal-priority BFS, and DFS. DFS is particularly harmful for regression ($R^2$: 16.8% vs. 25.2%), as deep-but-narrow traversal yields unbalanced contexts. Full BFS underperforms despite exploring more neighbors—under fixed token budgets, exhaustive low-hop expansion induces near-hop bias that crowds out informative long-range evidence.

Importantly, the main comparisons use the same sampling protocol for all pretrainable methods. To disentangle architectural gains from sampling effects, we additionally evaluate Griffin, RT, and RAMBA under Full BFS and DFS in Appendix B.3.7; RAMBA remains the best-performing model under all sampling variants, indicating that its gains are not solely due to the sampler.

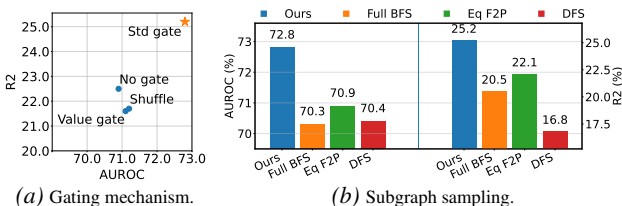

*(a)* Gating mechanism.      *(b)* Subgraph sampling.

*Figure 4.* **Ablation studies.**

### 6.4. Long-Context Scalability

A central motivation for RAMBA is enabling efficient processing of extensive relational contexts. We evaluate performance-efficiency trade-offs across context lengths $\{512, 1024, 2048, 4096\}$.

Figure 5 plots inference throughput against regression $R^2$. RAMBA occupies the Pareto-optimal region: it achieves the highest accuracy at each context length while maintaining substantially lower latency than RT. At 4096 tokens, RAMBA processes sequences **5.7× faster** than RT while

achieving **+3.7% higher** $R^2$.

RAMBA's performance *continues improving* with longer contexts (from 24.0% at 512 to 26.0% at 4096), while its latency grows only linearly. In contrast, RT's quadratic attention cost causes latency to surge at longer contexts. This confirms that $\mathcal{O}(L)$ complexity is not merely a computational convenience but a *prerequisite* for effectively leveraging rich contextual information in relational databases.

**Training efficiency and memory.** We further evaluate training throughput and peak memory consumption across context lengths. Although RT benefits from highly optimized attention kernels at short contexts, its quadratic cost becomes increasingly limiting as the context grows. In contrast, RAMBA scales more favorably: at 4096 tokens, RAMBA reaches 56 tasks/s compared with RT's 40 tasks/s, while using 29.1GB peak memory compared with RT's 37.3GB. These results clarify that our efficiency claim is not that RAMBA is uniformly faster at every sequence length, but that it provides better long-context scaling in both training throughput and memory usage. Full measurements are reported in Appendix D.3.

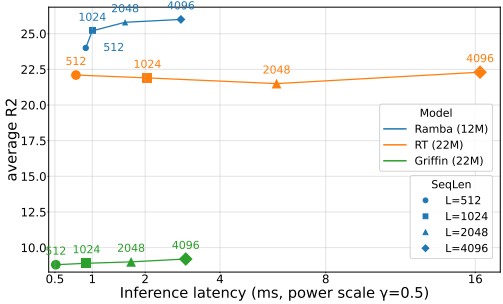

*Figure 5.* **Long-context scaling analysis.** Speed (tokens/sec) vs. Accuracy trade-off across context lengths.

### 6.5. Fine-Tuning Efficiency

We evaluate whether RAMBA's pretrained representations enable efficient adaptation with limited supervision, fine-tuning all models using only **1,024 labeled nodes per task**.

Figure 6 tracks performance across fine-tuning. RAMBA exhibits two key advantages: (1) *stronger initialization*—zero-shot performance (leftmost points) already exceeds most baselines' converged performance; and (2) *faster convergence*—RAMBA reaches 75% AUROC within 1k steps, while RT requires 10k+ steps to reach comparable levels.

For regression, RAMBA achieves 38.3% final $R^2$ compared to 26.8% for RT, representing a **43% relative improvement**. Task-specific baselines (RDL-GNN, RelGNN, RelGT) often yield negative $R^2$ in the few-shot regime, while RAMBA's pretrained representations provide robust initialization even for challenging numerical targets.

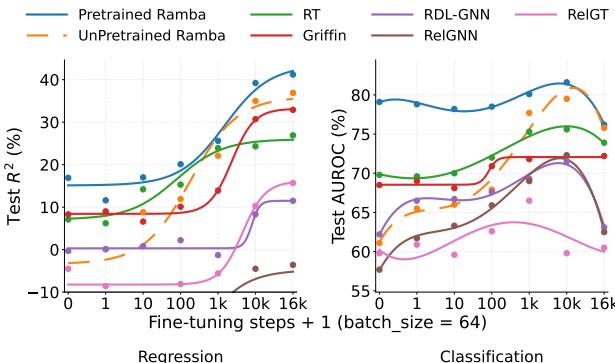

*Figure 6.* **Few-shot fine-tuning dynamics.** Test performance over training steps (log scale).

## 7. Conclusions and Limitations

We introduced RAMBA, the first selective state-space model designed for relational deep learning. By combining Topology-Aware Linearization with Schema Dynamic Gating, RAMBA resolves the tension between scalability and expressiveness: it processes vast relational contexts in linear time while selectively retaining semantically relevant information across tables. Our global columnar serialization preserves attribute-wise distributional patterns, while sparse structural attention recovers foreign-key topology without sacrificing efficiency. Experiments across diverse benchmarks demonstrate that RAMBA achieves state-of-the-art predictive performance while maintaining linear scalability.

Our current serialization strategy processes columns in a fixed order determined by the schema. While bidirectional scanning mitigates ordering effects, exploring adaptive or learned column orderings that prioritize more informative attributes may further enhance performance. We leave this investigation to future work.

### Acknowledgements

This work is supported in part by the Fundamental and Interdisciplinary Disciplines Breakthrough Plan of the Ministry of Education of China under contract No. JYB2025XDXM702, in part by Natural Science Foundation of China (NSFC) under contract No. 62506366 and 62572268, and in part by Ministry of Science and Technology of China under contract No. 2024YFB2809103.

### Impact Statement

This research addresses fundamental challenges in relational deep learning, including scalability limitations, cross-schema generalization, and the labor-intensive feature engineering bottleneck that currently impedes machine learning adoption on relational databases. By enabling efficient end-

to-end learning directly on multi-table database structures, RAMBA has the potential to democratize predictive analytics across domains such as e-commerce, healthcare, finance, and scientific research, where relational databases remain the dominant data storage paradigm.

No ethical concerns have been identified within the research methodology. The societal implications of this work are multifaceted: enabling organizations with limited machine learning expertise to leverage their existing relational data assets, reducing the substantial human effort currently required for manual feature engineering, and advancing the development of foundation models that can generalize across diverse database schemas without task-specific retraining. The linear scalability of our approach also contributes to computational efficiency and reduced energy consumption compared to quadratic-complexity alternatives, aligning with broader goals of sustainable AI development. Additionally, by learning directly from schema semantics rather than memorizing value distributions, RAMBA offers improved robustness to distribution shifts—a property increasingly important for deploying machine learning systems in dynamic real-world environments. These advancements collectively lower barriers to adoption and enhance the reliability of predictive modeling on the structured data that underlies much of modern enterprise and scientific infrastructure.

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

## A. RelBench Database Statistics and Design Motivation

To systematically characterize the structural properties of the RelBench databases (Robinson et al., 2024) and motivate our architectural choices, we conduct comprehensive analyses across seven datasets. This section provides: (i) task-level setup details, (ii) degree distribution analysis motivating width-limited BFS, (iii) column count statistics justifying sparse attention design, and (iv) subgraph size bounds illustrating scalability challenges.

### A.1. Task-Level Setup and Prediction Horizons

Table 5 summarizes all classification and regression tasks used in our experiments. Each task is characterized by its database domain, prediction horizon $\Delta$, and target definition. Notably, prediction horizons vary significantly both across and within databases—for instance, `rel-f1` includes tasks with $\Delta \in \{1\text{m}, 2\text{m}\}$—highlighting the heterogeneity of temporal dynamics in real-world relational data.

### A.2. Degree Distribution Analysis: Motivation for Width-Limited BFS

> **Key Insight: Heavy-Tailed Degree Distributions**
>
> Real-world relational databases exhibit *extreme heterogeneity* in node connectivity. While median out-degrees are typically small (1–2), maximum in-degrees can exceed $10^6$, creating "hub" nodes that cause explosive frontier growth during graph traversal.

Table 6 reports comprehensive degree statistics across all RelBench databases. Several critical observations emerge:

1. **Asymmetric degree distributions:** Out-degrees are bounded (typically $\leq 3$) due to the nature of foreign-key constraints, while in-degrees follow heavy-tailed distributions.

2. **Extreme hub nodes:** `rel-avito` contains nodes with in-degree exceeding 1.27M, while `rel-trial` and `rel-stack` have hubs with in-degrees $> 10^5$.

3. **Sparse connectivity on average:** Mean degrees range from 0.18 (`rel-event`) to 2.08 (`rel-avito`), indicating that most nodes have limited neighbors.

These statistics directly motivate our **width-limited BFS** strategy: unconstrained expansion from hub nodes would cause the sampling frontier to explode exponentially, overwhelming computational resources and injecting massive amounts of noise. Our width budget $N_{\text{child}}$ should be set to

*Table 5.* **Task-level overview of RelBench benchmarks.** Each row corresponds to a scalar-supervised task. $\Delta$ denotes the future prediction horizon (d=days, w=weeks, m=months, y=years). We report abstract target descriptions to avoid verbatim reproduction from the benchmark.

| DB | Domain | Task | Type / $\Delta$ | Target Description |
|---|---|---|---|---|
| rel-amazon | E-commerce reviews & purchases | user-churn | Cls / 3m | Binary indicator: zero reviews in future window |
| | | item-churn | Cls / 3m | Binary indicator: zero reviews received in future window |
| | | user-ltv | Reg / 3m | Aggregate monetary value from user activity |
| | | item-ltv | Reg / 3m | Aggregate monetary value from item transactions |
| rel-avito | Marketplace ads & search | user-visits | Cls / 4d | Threshold exceedance: distinct visited ads count |
| | | user-clicks | Cls / 4d | Threshold exceedance: distinct clicked ads count |
| | | ad-ctr | Reg / 4d | Click-through rate over prediction window |
| rel-event | Social events & RSVPs | user-attendance | Reg / 7d | Count of positive RSVPs in future window |
| | | user-repeat | Cls / 7d | Repeat participation indicator (conditioned on recent attendance) |
| | | user-ignore | Cls / 7d | Threshold exceedance: ignored invitations count |
| rel-f1 | Formula 1 racing | driver-dnf | Cls / 1m | Binary indicator: any DNF in future races |
| | | driver-top3 | Cls / 1m | Binary indicator: any top-3 qualifying position |
| | | driver-position | Reg / 2m | Mean finishing position across future races |
| rel-hm | Retail transactions | user-churn | Cls / 1w | Binary indicator: zero transactions in future window |
| | | item-sales | Reg / 1w | Aggregate transaction revenue for item |
| rel-stack | Q&A community | user-engagement | Cls / 3m | Binary indicator: any engagement activity |
| | | user-badge | Cls / 3m | Binary indicator: badge acquisition |
| | | post-votes | Reg / 3m | Vote accumulation count |
| rel-trial | Clinical trials | study-outcome | Cls / 1y | Binary primary endpoint success |
| | | study-adverse | Reg / 1y | Count of severe adverse events |
| | | site-success | Reg / 1y | Site-level success rate |

*Table 6.* **Node degree statistics across RelBench databases.** Out/In columns report minimum, maximum, and median values. The heavy-tailed in-degree distribution (note the extreme maxima) motivates width-limited BFS sampling.

| Dataset | Out-Degree | | | In-Degree | | | Mean |
|---|---|---|---|---|---|---|---|
| | Min | Max | Med | Min | Max | Med | |
| rel-amazon | 0 | 2 | 2 | 0 | 28,986 | 0 | 1.46 |
| rel-avito | 0 | 3 | 2 | 0 | **1,276,368** | 0 | 2.08 |
| rel-event | 0 | 2 | 0 | 0 | 914 | 0 | 0.18 |
| rel-f1 | 0 | 3 | 2 | 0 | 5,327 | 0 | 2.07 |
| rel-hm | 0 | 2 | 2 | 0 | 31,902 | 0 | 1.53 |
| rel-stack | 0 | 2 | 1 | 0 | 96,610 | 0 | 1.15 |
| rel-trial | 0 | 2 | 2 | 0 | 203,713 | 0 | 1.39 |

*Table 7.* **Per-table column count statistics.** Most tables have moderate column counts (mean $\approx$ 5–12), keeping the $K^2$ attention factor manageable. The outlier in rel-event (110 columns) represents a worst-case scenario.

| Dataset | Min | Max | Median | Mean |
|---|---|---|---|---|
| rel-amazon | 2 | 7 | 6 | 5.00 |
| rel-avito | 4 | 8 | 4.5 | 5.25 |
| rel-event | 3 | **110** | 6 | 26.20 |
| rel-f1 | 4 | 15 | 7 | 7.44 |
| rel-hm | 5 | 25 | 7 | 12.33 |
| rel-stack | 5 | 11 | 7 | 7.43 |
| rel-trial | 2 | 29 | 6 | 9.33 |

accommodate typical local expansion (guided by median degrees) rather than matching extreme in-degrees.

### A.3. Column Count Statistics: Implications for Attention Complexity

Our sparse structural attention operates within entity boundaries (intra-entity mask) and across foreign-key edges (relational mask). The computational cost includes a factor proportional to $K^2$, where $K$ is the number of columns per table. Table 7 characterizes this distribution.

The analysis reveals that:

- Mean column counts are modest (5–12 in most databases), making the $K^2$ factor computationally tractable.

- rel-event contains an extreme table with 110 columns, representing a potential bottleneck that our sparse masking efficiently handles.

*Table 8.* **Upper bounds on sampled subgraph sizes.** The heavy-tailed distribution (maxima reaching $10^7$, medians $\approx$ 3–78) illustrates why naive exhaustive sampling is impractical.

| DB | Min | Max | Median | Mean |
|---|---|---|---|---|
| rel-amazon | 1 | $4.0 \times 10^7$ | 11 | $3.1 \times 10^6$ |
| rel-avito | 1 | $2.5 \times 10^7$ | 3 | $5.0 \times 10^6$ |
| rel-event | 1 | $3.9 \times 10^5$ | 22 | $9.4 \times 10^3$ |
| rel-f1 | 1 | $1.2 \times 10^5$ | 2 | $4.1 \times 10^4$ |
| rel-hm | 1 | $2.4 \times 10^7$ | 58 | $3.5 \times 10^6$ |
| rel-stack | 1 | $1.1 \times 10^7$ | 78 | $7.0 \times 10^4$ |
| rel-trial | 4 | $6.1 \times 10^6$ | 23 | $1.6 \times 10^5$ |

- Our implementation leverages sparse/masked structures optimized for the typical column-count regime.

### A.4. Subgraph Size Bounds: Scalability vs. Noise Trade-off

Table 8 reports theoretical upper bounds on sampled subgraph sizes, revealing the fundamental tension between *scalability opportunity* and *noise management*.

> **Design Principle: Controlled Sampling Budget**
>
> While relational structures can theoretically provide massive context (up to $10^7$ nodes), naively expanding to such scales introduces overwhelming noise and computational cost. RAMBA's width-limited BFS combined with Schema Dynamic Gating enables *efficient extraction of useful evidence* within a controlled budget.

## B. Extended Experimental Results

### B.1. Statistical Significance over Five Seeds

To evaluate whether the observed gains are robust to random seeds, we repeat pretraining experiments with five random seeds and compare RAMBA against RT, the strongest zero-shot baseline in our main results. We report mean $\pm$ standard deviation and two-sided Welch's $t$-test in Tables 9 and 10. The results show that RAMBA's improvements are statistically significant on most classification and regression tasks, while also revealing task-level heterogeneity: not every task favors RAMBA, which is expected in RelBench due to diverse target semantics and database structures.

### B.2. Per-Task Long-Context Scaling Results

Table 11 provides complete per-task results across context lengths $\{512, 1024, 2048, 4096\}$, complementing the aggregated analysis in the main text.

**Key Observations.**

*Table 9.* Statistical significance on classification tasks over five seeds. We report AUROC (%).

| Dataset | Task | RT | RAMBA | $p$-value |
|---|---|---|---|---|
| rel-amazon | item-churn | $70.1 \pm 0.1$ | $\mathbf{71.0 \pm 0.2}$ | $< 0.001$ |
| rel-amazon | user-churn | $63.8 \pm 0.3$ | $\mathbf{64.5 \pm 0.5}$ | $0.033$ |
| rel-avito | user-clicks | $58.2 \pm 0.7$ | $\mathbf{59.3 \pm 0.3}$ | $0.021$ |
| rel-avito | user-visits | $61.2 \pm 0.2$ | $\mathbf{61.2 \pm 0.3}$ | $1.000$ |
| rel-f1 | driver-dnf | $76.9 \pm 0.3$ | $\mathbf{80.0 \pm 0.8}$ | $< 0.001$ |
| rel-f1 | driver-top3 | $\mathbf{89.2 \pm 0.2}$ | $87.6 \pm 0.4$ | $< 0.001$ |
| rel-hm | user-churn | $62.6 \pm 0.5$ | $\mathbf{69.4 \pm 0.1}$ | $< 0.001$ |
| rel-stack | user-badge | $\mathbf{79.2 \pm 0.2}$ | $77.9 \pm 0.3$ | $< 0.001$ |
| rel-stack | user-engage | $77.3 \pm 0.1$ | $\mathbf{90.7 \pm 0.8}$ | $< 0.001$ |
| rel-trial | study-out | $54.3 \pm 0.6$ | $\mathbf{61.4 \pm 0.3}$ | $< 0.001$ |

*Table 10.* Statistical significance on regression tasks over five seeds. We report $R^2$ (%).

| Dataset | Task | RT | RAMBA | $p$-value |
|---|---|---|---|---|
| rel-amazon | item-ltv | $32.9 \pm 0.4$ | $\mathbf{59.5 \pm 0.9}$ | $< 0.001$ |
| rel-amazon | user-ltv | $\mathbf{33.0 \pm 1.2}$ | $31.2 \pm 1.7$ | $0.093$ |
| rel-avito | ad-ctr | $4.5 \pm 0.1$ | $\mathbf{8.5 \pm 0.9}$ | $< 0.001$ |
| rel-f1 | driver-pos | $\mathbf{55.0 \pm 0.6}$ | $47.3 \pm 0.6$ | $< 0.001$ |
| rel-hm | item-sales | $13.0 \pm 0.8$ | $\mathbf{13.6 \pm 1.1}$ | $0.355$ |
| rel-stack | post-votes | $31.2 \pm 1.0$ | $\mathbf{34.0 \pm 1.0}$ | $0.002$ |
| rel-trial | site-succ | $2.6 \pm 0.4$ | $\mathbf{5.6 \pm 0.3}$ | $< 0.001$ |
| rel-trial | study-adv | $-0.0 \pm 0.1$ | $\mathbf{1.2 \pm 0.1}$ | $< 0.001$ |

1. **RAMBA benefits from longer contexts:** Mean regression $R^2$ improves from 24.0% (512 tokens) to 26.0% (4096 tokens), a consistent upward trend.

2. **Task-specific sensitivity:** Some tasks (e.g., rel-amazon/item-ltv) show dramatic improvement with longer contexts (+10.3% $R^2$ from 512 to 4096), while others are relatively stable.

3. **RT struggles with long contexts:** Unlike RAMBA, RT does not consistently benefit from longer sequences, with mean performance actually degrading slightly at 4096 tokens for regression.

4. **Griffin underperforms:** Despite sharing linear complexity, Griffin achieves substantially lower scores across all context lengths, indicating that RAMBA's architectural innovations (Schema Dynamic Gating, Sparse Structural Attention) provide crucial benefits beyond efficient scaling.

### B.3. Detailed Ablation Results

We provide complete per-task breakdowns for all ablation studies presented in the main text.

#### B.3.1. MODULE ABLATION

Table 12 reports results when selectively enabling/disabling the Mamba backbone (`mamba`), intra-entity attention (`int`), and relational attention (`rel`).

*Table 11.* **Per-task long-context scaling results.** Performance of RAMBA, RT, and Griffin across context lengths. Upper block: classification (AUROC %); lower block: regression ($R^2$ %). **Bold**: best per task-length combination.

| Dataset | Task | RAMBA | | | | RT | | | | Griffin | | | |
|---|---|---|---|---|---|---|---|---|---|---|---|---|---|
| | | 512 | 1024 | 2048 | 4096 | 512 | 1024 | 2048 | 4096 | 512 | 1024 | 2048 | 4096 |
| **Classification (AUROC %)** | | | | | | | | | | | | | |
| rel-amazon | item-churn | 72.5 | **73.0** | **74.1** | **72.3** | **75.6** | 70.2 | 70.3 | 69.9 | 68.3 | 69.0 | 69.5 | 68.9 |
| rel-amazon | user-churn | **65.5** | **65.0** | **65.1** | **64.1** | 64.0 | 63.9 | 64.1 | 63.0 | 60.9 | 62.3 | 62.1 | 61.5 |
| rel-avito | user-clicks | **60.0** | **59.5** | 54.4 | **63.7** | 44.4 | 58.0 | **55.9** | 57.9 | 45.4 | 45.9 | 46.6 | 46.0 |
| rel-avito | user-visits | 58.8 | 61.2 | 59.0 | 59.8 | 57.0 | **61.4** | 58.0 | **62.2** | **61.1** | 60.7 | **62.3** | 61.7 |
| rel-f1 | driver-dnf | **81.3** | **79.7** | **80.0** | **80.0** | 78.4 | 77.2 | 78.6 | 76.2 | 56.8 | 57.7 | 57.4 | 56.8 |
| rel-f1 | driver-top3 | 87.4 | 88.3 | **88.4** | **89.3** | **88.2** | **89.1** | 87.3 | 88.1 | 81.0 | 82.5 | 82.2 | 81.6 |
| rel-hm | user-churn | 68.0 | **69.1** | 67.4 | **68.6** | **68.8** | 62.8 | 63.5 | 63.7 | 60.7 | 60.2 | 61.9 | 61.3 |
| rel-stack | user-badge | **78.7** | 76.7 | **79.6** | 78.5 | **78.3** | **79.0** | 75.6 | 79.0 | 73.1 | 73.5 | 74.3 | 73.7 |
| rel-stack | user-engage | **92.3** | **93.8** | **92.8** | **92.6** | 90.0 | 77.1 | 83.6 | 75.8 | 75.8 | 77.5 | 77.0 | 76.4 |
| rel-trial | study-out | **55.8** | **61.6** | **60.0** | **59.2** | 52.1 | 54.5 | 56.4 | 54.3 | 50.4 | 51.0 | 51.6 | 51.0 |
| **Mean** | | **72.0** | **72.8** | **72.0** | **72.8** | 69.7 | 69.3 | 69.3 | 69.0 | 63.3 | 64.0 | 64.5 | 63.9 |
| **Regression ($R^2$ %)** | | | | | | | | | | | | | |
| rel-amazon | item-ltv | **51.7** | **54.0** | **56.4** | **62.0** | 49.0 | 33.2 | 43.6 | 33.5 | 19.9 | 20.1 | 20.1 | 20.3 |
| rel-amazon | user-ltv | **34.2** | 33.1 | **34.2** | **35.8** | 31.9 | **33.9** | 28.6 | 33.1 | 19.7 | 20.6 | 19.9 | 20.1 |
| rel-avito | ad-ctr | **9.0** | **9.3** | 7.3 | **9.4** | 8.8 | 4.5 | 5.5 | 5.1 | 2.4 | 2.4 | 2.6 | 2.8 |
| rel-f1 | driver-pos | 44.3 | 47.8 | 47.4 | 46.6 | **54.6** | **54.7** | 50.8 | **56.9** | 0.2 | -0.7 | 0.4 | 0.6 |
| rel-hm | item-sales | **13.0** | **14.1** | **17.2** | **17.2** | 3.9 | 14.0 | 5.8 | 13.0 | 1.7 | 2.7 | 1.9 | 2.1 |
| rel-stack | post-votes | **33.9** | **36.3** | **38.0** | **32.1** | 30.6 | 32.2 | 28.2 | 31.2 | 26.4 | 27.4 | 26.6 | 26.8 |
| rel-trial | site-succ | 4.3 | **5.7** | **5.0** | **5.0** | **5.3** | 2.7 | 0.3 | **5.0** | 2.4 | 1.4 | 2.6 | 2.8 |
| rel-trial | study-adv | **2.0** | **1.3** | 0.5 | 0.0 | 0.1 | -0.1 | **0.8** | **0.7** | -2.4 | -2.5 | -2.2 | -2.0 |
| **Mean** | | **24.0** | **25.2** | **25.7** | **26.0** | 22.1 | 21.9 | 21.5 | 22.3 | 8.8 | 8.9 | 9.0 | 9.2 |

**Analysis.**

- **Mamba alone** provides substantial gains (+18.5% AU-ROC, +8.2% $R^2$) over the baseline, demonstrating the value of global columnar context.

- **Attention without Mamba** (int+rel) achieves competitive classification (72.8% AUROC) but struggles with regression (17.5% $R^2$).

- **Full model synergy**: The complete architecture achieves the best regression performance (25.2% $R^2$), confirming that both components contribute complementary information.

#### B.3.2. SERIALIZATION STRATEGY ABLATION

Table 13 compares different serialization orderings.

#### B.3.3. COLUMN-ORDER RANDOMIZATION

We further test whether RAMBA depends on a brittle fixed column order. In addition to the serialization ablations in Table 13, we evaluate two stronger perturbations: *Col-block random*, which randomly permutes column blocks, and *Within-col random*, which shuffles cells within each column block. Table 14 and Table 15 show that both perturbations

lead to only small performance changes, suggesting that RAMBA's gains come primarily from column-level distributional grouping and schema-aware filtering rather than memorizing a fixed schema order.

#### B.3.4. SCHEMA DYNAMIC GATING ABLATION

Table 16 evaluates different gating mechanisms.

> **Critical Finding: Schema Semantics Matter**
>
> The Shuffle control performs *worse* than No gate, demonstrating that incorrect semantic alignment is more harmful than no alignment at all. This confirms that the gating mechanism genuinely leverages schema semantics rather than learning spurious correlations.

#### B.3.5. ROBUSTNESS TO SCHEMA NAMING DEGRADATION

A potential concern is that Schema Dynamic Gating may rely too heavily on descriptive column names. To evaluate this, we systematically degrade schema names while keeping the remaining model and evaluation protocol unchanged. We consider three degraded settings: *Generic names*, where

*Table 12.* **Per-task module ablation results.** `none`: task head only; `all`: full model. We report AUROC (%) for classification and $R^2$ (%) for regression.

| Dataset | Task | none | mamba | int | int+mamba | rel | rel+mamba | int+rel | all |
|---|---|---|---|---|---|---|---|---|---|
| | | | | **Classification (AUROC %)** | | | | | |
| rel-amazon | item-churn | 50.0 | 72.1 | 54.5 | 72.0 | 50.2 | 71.8 | **73.2** | 73.0 |
| rel-amazon | user-churn | 50.0 | 63.8 | 51.6 | 64.9 | 47.1 | 64.5 | **65.5** | 65.0 |
| rel-avito | user-clicks | 50.0 | 55.9 | 54.2 | 57.8 | 51.0 | 58.5 | **67.1** | 59.5 |
| rel-avito | user-visits | 50.0 | 57.6 | 53.0 | 57.8 | 55.4 | 60.1 | 58.0 | **61.2** |
| rel-f1 | driver-dnf | 50.0 | 74.0 | 56.2 | **79.9** | 46.6 | 77.3 | 79.2 | 79.7 |
| rel-f1 | driver-top3 | 50.0 | 83.2 | 57.5 | **89.6** | 46.6 | 85.9 | 89.1 | 88.3 |
| rel-hm | user-churn | 50.0 | 64.9 | 58.2 | 67.2 | 55.7 | 67.4 | 66.4 | **69.1** |
| rel-stack | user-badge | 50.0 | 68.4 | 54.5 | 74.0 | 51.6 | 75.3 | **77.4** | 76.7 |
| rel-stack | user-engage | 50.0 | 87.0 | 50.2 | 92.1 | 45.4 | 90.6 | 88.9 | **93.8** |
| rel-trial | study-out | 50.0 | 58.1 | 60.1 | 60.9 | 50.5 | 60.6 | **63.2** | 61.6 |
| **Mean** | | 50.0 | 68.5 | 55.0 | 71.6 | 50.0 | 71.2 | 72.8 | **72.8** |
| | | | | **Regression ($R^2$ %)** | | | | | |
| rel-amazon | item-ltv | 0.0 | 6.8 | 0.7 | 42.8 | 0.0 | 14.8 | 51.5 | **54.0** |
| rel-amazon | user-ltv | 0.0 | 17.9 | 0.0 | 23.2 | -0.1 | 21.9 | 30.9 | **33.1** |
| rel-avito | ad-ctr | 0.0 | -4.6 | 0.0 | 2.8 | 0.0 | 3.3 | 4.1 | **9.3** |
| rel-f1 | driver-pos | 0.0 | 32.7 | -3.1 | 40.6 | 0.1 | 37.8 | 40.3 | **47.8** |
| rel-hm | item-sales | 0.0 | 1.4 | 0.1 | 6.2 | -0.1 | 1.9 | 2.6 | **14.1** |
| rel-stack | post-votes | 0.0 | 32.3 | 1.3 | 27.6 | -0.1 | 32.7 | 25.0 | **36.3** |
| rel-trial | site-succ | 0.0 | -8.4 | -0.3 | -2.0 | 0.1 | -0.5 | -4.9 | **5.7** |
| rel-trial | study-adv | 0.0 | -12.5 | -1.9 | -5.2 | 0.0 | -4.6 | -9.5 | **1.3** |
| **Mean** | | 0.0 | 8.2 | -0.4 | 17.0 | 0.0 | 13.4 | 17.5 | **25.2** |

columns are replaced by uninformative identifiers such as `col_1`; *Random names*, where column names are randomly reassigned; and *No names*, where column-name information is removed. Table 17 shows that RAMBA degrades gracefully rather than collapsing. Classification is only mildly affected, while regression is more sensitive to schema semantics, as expected.

### B.3.6. SAMPLING STRATEGY ABLATION

Table 18 compares different graph sampling strategies.

### B.3.7. SAMPLING VARIANTS ACROSS METHODS

The main results use the same width-limited BFS sampler with foreign-key prioritization for all pretrainable methods. To separate the effect of the sampler from the effect of the architecture, we additionally evaluate Griffin, RT, and RAMBA under alternative sampling strategies. Table 19 reports average zero-shot AUROC and $R^2$ under our sampler, Full BFS, and DFS. While our sampler improves the evidence distribution for all methods, RAMBA remains the strongest model under all three sampling variants. This suggests that the gains arise from the interaction between high-quality relational context and RAMBA's architecture, rather than from an unfair sampling advantage.

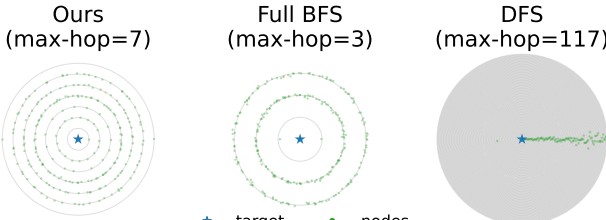

*Figure 7.* **Sampled subgraph visualization.** Left: width-limited BFS (max-hop=7); Middle: full BFS (max-hop=3); Right: DFS (max-hop=117). Width-limited BFS achieves balanced coverage.

### B.4. Sampled Subgraph Examples

Figure 7 visualizes subgraphs from different sampling strategies. Width-limited BFS produces balanced coverage with moderate depth, reaching distant nodes while maintaining coherent neighborhoods.

### B.5. Gating Visualization

Figure 8 visualizes the learned $\Delta$ modulation for predicting `review_rating`. The gating mechanism exhibits clear column-level structure: semantically relevant columns (`review_rating`, `product_price`) receive high $\Delta$ values, while weakly related columns (`product_category`, `product_brand`) are sup-

*Table 13.* **Per-task serialization ablation.** `Random`: global permutation; `Row`: row-wise order; `Col Shuf`: column blocks with within-block shuffling; `Col`: column-wise (default).

| Dataset | Task | Random | Row | Col Shuf | Col |
|---------|------|--------|-----|----------|-----|
| **Classification (AUROC %)** | | | | | |
| rel-amazon | item-churn | 71.3 | 71.6 | 71.0 | **73.0** |
| rel-amazon | user-churn | 64.7 | **65.0** | 64.8 | **65.0** |
| rel-avito | user-clicks | **59.8** | 58.6 | 58.6 | 59.5 |
| rel-avito | user-visits | 53.8 | 60.2 | 59.9 | **61.2** |
| rel-f1 | driver-dnf | 78.8 | 79.1 | **80.1** | 79.7 |
| rel-f1 | driver-top3 | 87.2 | 86.1 | **89.0** | 88.3 |
| rel-hm | user-churn | 66.3 | 68.4 | 66.8 | **69.1** |
| rel-stack | user-badge | 76.2 | **78.1** | 75.1 | 76.7 |
| rel-stack | user-engage | 90.6 | 91.3 | 92.8 | **93.8** |
| rel-trial | study-out | 57.5 | 55.6 | 58.0 | **61.6** |
| **Mean** | | 70.6 | 71.4 | 71.2 | **72.8** |
| **Regression ($R^2$ %)** | | | | | |
| rel-amazon | item-ltv | 60.2 | 56.1 | **61.0** | 54.0 |
| rel-amazon | user-ltv | 27.0 | 31.6 | 28.3 | **33.1** |
| rel-avito | ad-ctr | 6.2 | 7.2 | 8.1 | **9.3** |
| rel-f1 | driver-pos | 42.2 | 43.7 | 47.0 | **47.8** |
| rel-hm | item-sales | 4.9 | 8.3 | 12.7 | **14.1** |
| rel-stack | post-votes | 26.1 | 26.2 | 30.7 | **36.3** |
| rel-trial | site-succ | 3.7 | 3.8 | 4.8 | **5.7** |
| rel-trial | study-adv | 0.0 | 0.2 | 0.2 | **1.3** |
| **Mean** | | 21.3 | 22.1 | 24.0 | **25.2** |

*Table 14.* Column-order randomization on classification tasks. We report AUROC (%).

| Task | Col-block random | Within-col random | Default |
|------|------------------|-------------------|---------|
| item-churn | 71.7 | 64.8 | 71.1 |
| user-churn (Amazon) | 64.0 | 71.0 | 64.9 |
| user-clicks | 59.9 | 58.6 | 59.5 |
| user-visits | 60.5 | 59.9 | 61.2 |
| driver-dnf | 80.2 | 80.1 | 80.5 |
| driver-top3 | 87.5 | 89.0 | 87.4 |
| user-churn (HM) | 69.0 | 66.8 | 69.4 |
| Average | 70.4 | 70.0 | **70.6** |

*Table 15.* Column-order randomization on regression tasks. We report $R^2$ (%).

| Task | Col-block random | Within-col random | Default |
|------|------------------|-------------------|---------|
| item-ltv | 61.0 | 61.0 | 61.4 |
| user-ltv | 34.1 | 28.3 | 33.7 |
| ad-ctr | 8.9 | 8.1 | 9.3 |
| driver-pos | 47.8 | 47.0 | 48.2 |
| item-sales | 12.7 | 12.7 | 14.1 |
| Average | 32.9 | 31.4 | **33.3** |

pressed. This validates that Schema Dynamic Gating identifies task-relevant information based on semantic alignment.

## C. Detailed Methodology

This section provides comprehensive technical details on RAMBA's components, complementing the high-level description in the main text.

### C.1. Cell Embedding Details

Each cell $\xi = (v, c, \tau, T)$ is embedded through a fusion of schema and value information. We detail each component below.

**Schema Embedding.** We leverage a frozen sentence transformer (specifically, `all-MiniLM-L12-v2` with $D_{\text{LM}} = 384$) to encode column semantics:

$$\mathbf{e}_{\text{schema}} = \phi_{\text{LM}}(\texttt{concat}(T, \texttt{":\ \ "}, c)) \in \mathbb{R}^{D_{\text{LM}}} \quad (16)$$

This produces semantically meaningful representations that capture both table context and column semantics, enabling cross-table transfer.

**Value Embeddings by Type.** We use type-specific encoders on preprocessed inputs:

- **Numerical** ($\tau = \texttt{num}$): Z-score normalization plus linear projection:

$$v_{\text{norm}} = \frac{v - \mu}{\sigma} \quad (17)$$

$$\mathbf{e}_{\text{value}} = \text{RMSNorm}\big(\mathbf{W}_{\text{num}}[v_{\text{norm}}] + \mathbf{b}_{\text{num}}\big) \in \mathbb{R}^{D} \quad (18)$$

  where $\mu, \sigma$ are column-wise mean and standard deviation from preprocessing, and $\mathbf{W}_{\text{num}} \in \mathbb{R}^{D \times 1}$.

- **Categorical / text** ($\tau = \texttt{text}$): Frozen language-model embeddings plus linear projection. Each distinct string is mapped to an index $v \in \{0, \ldots, |\mathcal{V}| - 1\}$; the preprocessing stage computes $\mathbf{h}(v) \in \mathbb{R}^{D_{\text{text}}}$ with a frozen sentence encoder, then:

$$\mathbf{e}_{\text{value}} = \text{RMSNorm}\big(\mathbf{W}_{\text{text}}\, \mathbf{h}(v) + \mathbf{b}_{\text{text}}\big) \in \mathbb{R}^{D} \quad (19)$$

  where $\mathbf{W}_{\text{text}} \in \mathbb{R}^{D \times D_{\text{text}}}$.

- **Temporal** ($\tau = \texttt{temp}$): Z-score normalized timestamp plus linear projection:

$$t_{\text{norm}} = \frac{t - \mu_t}{\sigma_t}, \quad (20)$$

$$\mathbf{e}_{\text{value}} = \text{RMSNorm}\big(\mathbf{W}_{\text{temp}}[t_{\text{norm}}] + \mathbf{b}_{\text{temp}}\big) \in \mathbb{R}^{D} \quad (21)$$

  with column-wise $\mu_t, \sigma_t$ from preprocessing.

- **Boolean**: Scalar value (normalized or 0/1) plus linear projection:

$$\mathbf{e}_{\text{value}} = \text{RMSNorm}\big(\mathbf{W}_{\text{bool}}[v] + \mathbf{b}_{\text{bool}}\big) \in \mathbb{R}^{D}. \quad (22)$$

*Table 16.* **Per-task gating ablation.** `No gate`: gating disabled; `Value gate`: value-based gating; `Shuffle`: randomized schema signals (control); `Std gate`: schema-based gating (default).

| Dataset | Task | No gate | Value gate | Shuffle | Std gate |
|---|---|---|---|---|---|
| **Classification (AUROC %)** | | | | | |
| rel-amazon | item-churn | 72.5 | 72.4 | 72.9 | **73.0** |
| rel-amazon | user-churn | **65.0** | 64.9 | 64.8 | **65.0** |
| rel-avito | user-clicks | 57.2 | 58.2 | 58.3 | **59.5** |
| rel-avito | user-visits | 52.8 | 52.8 | 55.7 | **61.2** |
| rel-f1 | driver-dnf | 79.3 | 79.2 | 78.4 | **79.7** |
| rel-f1 | driver-top3 | 87.8 | 87.3 | **89.4** | 88.3 |
| rel-hm | user-churn | 67.5 | 67.7 | 67.4 | **69.1** |
| rel-stack | user-badge | 77.3 | **79.3** | 76.6 | 76.7 |
| rel-stack | user-engage | 92.6 | 90.9 | 90.8 | **93.8** |
| rel-trial | study-out | 56.9 | 58.0 | 57.7 | **61.6** |
| **Mean** | | 70.9 | 71.1 | 71.2 | **72.8** |
| **Regression ($R^2$ %)** | | | | | |
| rel-amazon | item-ltv | 53.1 | 53.6 | **55.9** | 54.0 |
| rel-amazon | user-ltv | 32.5 | 30.0 | 29.9 | **33.1** |
| rel-avito | ad-ctr | 8.0 | 5.7 | 5.7 | **9.3** |
| rel-f1 | driver-pos | 46.8 | 44.9 | 44.2 | **47.8** |
| rel-hm | item-sales | 4.9 | 3.0 | 7.5 | **14.1** |
| rel-stack | post-votes | 31.5 | 33.7 | 27.6 | **36.3** |
| rel-trial | site-succ | 3.0 | 2.2 | 2.8 | **5.7** |
| rel-trial | study-adv | 0.2 | 0.0 | -0.2 | **1.3** |
| **Mean** | | 22.5 | 21.6 | 21.7 | **25.2** |

*Table 17.* Robustness to schema naming degradation. We report average zero-shot AUROC for classification and $R^2$ for regression.

| Schema naming setting | AUROC (%) | $R^2$ (%) |
|---|---|---|
| Original names | **72.5** | **25.9** |
| Generic names | 71.4 | 21.8 |
| Random names | 71.0 | 22.8 |
| No names | 70.9 | 22.5 |

*Table 18.* **Per-task sampling strategy ablation.** `DFS`: depth-first search; `Eq F2P`: equal priority BFS; `Full BFS`: unconstrained BFS; `Ours`: width-limited BFS with FK prioritization.

| Dataset | Task | DFS | Eq F2P | Full BFS | Ours |
|---|---|---|---|---|---|
| **Classification (AUROC %)** | | | | | |
| rel-amazon | item-churn | 72.6 | **73.6** | 72.1 | 73.0 |
| rel-amazon | user-churn | 64.6 | **65.3** | 64.8 | 65.0 |
| rel-avito | user-clicks | 57.3 | 58.9 | 57.9 | **59.5** |
| rel-avito | user-visits | 52.8 | 53.1 | 53.6 | **61.2** |
| rel-f1 | driver-dnf | 76.7 | 79.3 | 78.4 | **79.7** |
| rel-f1 | driver-top3 | 85.9 | 87.8 | 85.9 | **88.3** |
| rel-hm | user-churn | 68.1 | 67.1 | 67.3 | **69.1** |
| rel-stack | user-badge | **79.2** | 76.4 | 75.3 | 76.7 |
| rel-stack | user-engage | 90.8 | 91.9 | 91.0 | **93.8** |
| rel-trial | study-out | 55.5 | 55.4 | 56.6 | **61.6** |
| **Mean** | | 70.4 | 70.9 | 70.3 | **72.8** |
| **Regression ($R^2$ %)** | | | | | |
| rel-amazon | item-ltv | 46.5 | 53.0 | 47.8 | **54.0** |
| rel-amazon | user-ltv | 29.3 | 27.4 | 30.7 | **33.1** |
| rel-avito | ad-ctr | 6.0 | 6.5 | 5.4 | **9.3** |
| rel-f1 | driver-pos | 18.4 | **47.9** | 42.2 | 47.8 |
| rel-hm | item-sales | 3.0 | 5.9 | 2.9 | **14.1** |
| rel-stack | post-votes | 26.7 | 32.0 | 32.2 | **36.3** |
| rel-trial | site-succ | 4.0 | 4.2 | 2.9 | **5.7** |
| rel-trial | study-adv | 0.7 | 0.0 | 0.0 | **1.3** |
| **Mean** | | 16.8 | 22.1 | 20.5 | **25.2** |

*Table 19.* Sampling variants across methods. We report average zero-shot AUROC (%) and $R^2$ (%).

| Method | Ours | | Full BFS | | DFS | |
|---|---|---|---|---|---|---|
| | AUROC | $R^2$ | AUROC | $R^2$ | AUROC | $R^2$ |
| Griffin | 64.0 | 8.9 | 63.6 | 2.0 | 62.1 | 2.3 |
| RT | 69.3 | 21.9 | 70.0 | 18.1 | 69.3 | 13.7 |
| Ramba | **72.5** | **25.9** | 70.3 | 20.5 | 70.4 | 16.8 |

**Schema (column-name) embedding.** Column names are encoded with the same frozen language model as text values, then projected:

$$\mathbf{e}_{\text{schema}} = \text{RMSNorm}\big(\mathbf{W}_{\text{col}}\,\mathbf{h}_{\text{col}}(c) + \mathbf{b}_{\text{col}}\big) \in \mathbb{R}^D, \quad (23)$$

where $c$ indexes the column and $\mathbf{h}_{\text{col}}(c) \in \mathbb{R}^{D_{\text{text}}}$ is the frozen embedding of the column name.

**Scale embedding (preprocessing).** Scale is computed in preprocessing from the magnitude of the raw value relative to the column. For numerical and temporal types we use

$$s = \tanh\left(\log_{10}\left(\frac{|v - \mu| + \epsilon}{9}\right)\right) \in [-1, 1], \quad (24)$$

with $\epsilon = 10^{-8}$ and column-wise $\mu$; for text, $s = 0$. This $s$ is passed as `scale_values` into the model.

**Final cell embedding.** The cell embedding is the sum of schema, scale, and value (or mask) terms; the current implementation does not use a separate table-position embedding:

$$\mathbf{x}_\xi = \underbrace{\mathbf{e}_{\text{schema}}}_{\text{schema}} + \underbrace{\alpha \cdot \tanh\big(\mathbf{W}_{\text{scale}}\,[s] + \mathbf{b}_{\text{scale}}\big)}_{\text{scale}} + \underbrace{\mathbf{e}_{\text{value}}}_{\text{value}}$$

(25)

where $\tau \in \{\texttt{num}, \texttt{text}, \texttt{temp}, \texttt{bool}\}$ is the semantic type of the cell, $\mathbf{v}_\tau$ is the preprocessed input (scalar or $\mathbf{h}(v)$ as above), and $\alpha \in \mathbb{R}$ is a learnable scalar initialized to 0.1. For masked cells, $\mathbf{e}_{\text{value}}$ is replaced by a learnable type-specific mask vector $\mathbf{m}_\tau \in \mathbb{R}^D$. The scale term is applied only at non-masked, non-padding positions.

### C.2. Subgraph Sampling Algorithm

Algorithm 1 presents our width-limited BFS sampling procedure with foreign-key prioritization.

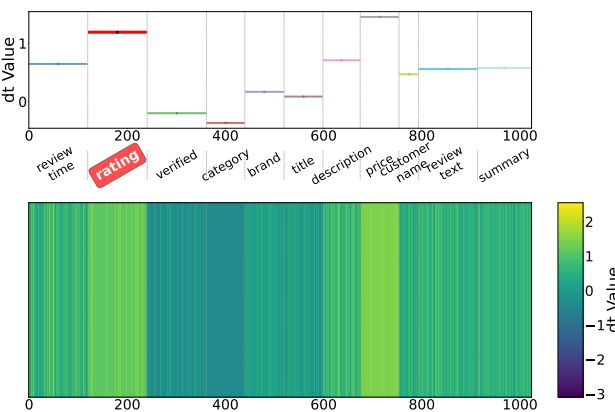

*Figure 8.* **Learned gating modulation.** Bottom: $\Delta$ values across sequence positions; Top: column-wise means. Target: `review_rating`. Semantically relevant columns receive higher $\Delta$, while weakly related columns are suppressed.

1. **FK Prioritization:** Foreign-key parents are always included (line 6–10), ensuring relational integrity is preserved.

2. **Width Budget:** Children are sampled with budget $N_{\text{child}}$ (line 12), preventing hub-induced explosion.

3. **Temporal Filtering:** Only children within the lookback window and satisfying causality ($t(r') \leq t(r^*)$) are considered (lines 11, 14).

### C.3. Schema Dynamic Gating: Mathematical Details

The Schema Dynamic Gating mechanism modulates SSM state transitions based on semantic relevance. We provide the complete mathematical formulation.

**Semantic Context Computation.** For target attribute $c^*$ and current scanning position in column $c_k$:

$$\mathbf{s}_{\text{tgt}} = \phi_{\text{LM}}(\texttt{desc}(c^*)) \in \mathbb{R}^{D_{\text{LM}}} \tag{26}$$

$$\mathbf{s}_{\text{curr}}(t) = \phi_{\text{LM}}(\texttt{desc}(c_k)) \in \mathbb{R}^{D_{\text{LM}}}, \quad \text{if } x_t \in \mathcal{C}_k \tag{27}$$

where $\texttt{desc}(\cdot)$ concatenates table name and column name with appropriate formatting.

**Relevance Gate Computation.** The gate score is computed via a two-layer MLP:

$$g_t^{\text{sem}} = \sigma\left(\mathbf{w}_g^\top \cdot \text{GELU}\left(\mathbf{W}_{\text{rel}}\begin{bmatrix}\mathbf{s}_{\text{curr}}(t) \\ \mathbf{s}_{\text{tgt}}\end{bmatrix}\right) + b_g\right) \in (0,1) \tag{28}$$

where:

- $\mathbf{W}_{\text{rel}} \in \mathbb{R}^{D_g \times 2D_{\text{LM}}}$ projects the concatenated semantic contexts

- $\mathbf{w}_g \in \mathbb{R}^{D_g}$ computes the final scalar relevance score

- $\sigma(\cdot)$ is the sigmoid function ensuring $g_t^{\text{sem}} \in (0,1)$

---

**Algorithm 1** Width-Limited BFS Sampling with FK Prioritization

**Require:** Target node $r^*$, max depth $K$, child budget $N_{\text{child}}$, lookback window $\Delta\tau$
**Ensure:** Sampled subgraph $\mathcal{G}_{\text{sub}} = (\mathcal{V}_{\text{sub}}, \mathcal{E}_{\text{sub}})$

1: $\mathcal{V}_{\text{sub}} \leftarrow \{r^*\}$, $\mathcal{E}_{\text{sub}} \leftarrow \emptyset$
2: $\text{frontier}^{\text{FK}} \leftarrow [r^*]$, $\text{frontier}^{\text{child}} \leftarrow \emptyset$, $\text{depth} \leftarrow 0$
3: **while** ($\text{frontier}^{\text{FK}} \neq \emptyset$ **or** $\text{frontier}^{\text{child}} \neq \emptyset$) **and** $\text{depth} < K$ **do**
4:     *// Priority: expand FK-parent stack first, then child stack*
5:     **if** $\text{frontier}^{\text{FK}} \neq \emptyset$ **then**
6:         $r \leftarrow \text{POP}(\text{frontier}^{\text{FK}})$
7:     **else**
8:         $r \leftarrow \text{POP}(\text{frontier}^{\text{child}})$
9:     **end if**
10:    *// Process foreign-key parents (full inclusion)*
11:    **for each** parent $r'$ via FK reference from $r$ **do**
12:       $\mathcal{V}_{\text{sub}} \leftarrow \mathcal{V}_{\text{sub}} \cup \{r'\}$
13:       $\mathcal{E}_{\text{sub}} \leftarrow \mathcal{E}_{\text{sub}} \cup \{(r, r')\}$
14:       $\text{frontier}^{\text{FK}} \leftarrow \text{frontier}^{\text{FK}} \cup \{r'\}$
15:    **end for**
16:    *// Process children (time-restricted, budgeted)*
17:    $\text{children} \leftarrow \{r' : (r', r) \in \mathcal{E},\ t(r') \in [t(r^*) - \Delta\tau, t(r^*)]\}$
18:    $\text{sampled} \leftarrow \text{SAMPLE}(\text{children}, \min(|\text{children}|, N_{\text{child}}))$
19:    **for each** $r' \in \text{sampled}$ **do**
20:       **if** $t(r') \leq t(r^*)$ **then**
21:         $\mathcal{V}_{\text{sub}} \leftarrow \mathcal{V}_{\text{sub}} \cup \{r'\}$
22:         $\mathcal{E}_{\text{sub}} \leftarrow \mathcal{E}_{\text{sub}} \cup \{(r', r)\}$
23:         $\text{frontier}^{\text{child}} \leftarrow \text{frontier}^{\text{child}} \cup \{r'\}$
24:       **end if**
25:    **end for**
26:    $\text{depth} \leftarrow \text{depth} + 1$
27: **end while**
28: **Return** $(\mathcal{V}_{\text{sub}}, \mathcal{E}_{\text{sub}})$

---

**Modulated State Dynamics.** The gate modulates the discretization step $\Delta_t$:

$$\tilde{\Delta}_t = \underbrace{\text{softplus}(\mathbf{W}_\Delta \mathbf{x}_t + \mathbf{b}_\Delta)}_{\text{base step}} \odot \underbrace{g_t^{\text{sem}}}_{\text{semantic gate}} \tag{29}$$

$$\tilde{\bar{\mathbf{A}}}_t = \exp(\text{diag}(\tilde{\Delta}_t)\mathbf{A}) \tag{30}$$

$$\tilde{\bar{\mathbf{B}}}_t = \text{diag}(\tilde{\Delta}_t)\mathbf{B}_t \tag{31}$$

**Gating Behavior Analysis**

- When $g_t^{\text{sem}} \to 1$ (high relevance): $\tilde{\Delta}_t \approx \Delta_t$, normal state evolution occurs, and information is integrated into the hidden state.

- When $g_t^{\text{sem}} \to 0$ (low relevance): $\tilde{\Delta}_t \to \mathbf{0}$, causing $\tilde{\bar{\mathbf{A}}}_t \to \mathbf{I}$ and $\tilde{\bar{\mathbf{B}}}_t \to \mathbf{0}$. The state passes unchanged: $\mathbf{h}_t \approx \mathbf{h}_{t-1}$, effectively "freezing" memory and bypassing irrelevant information.

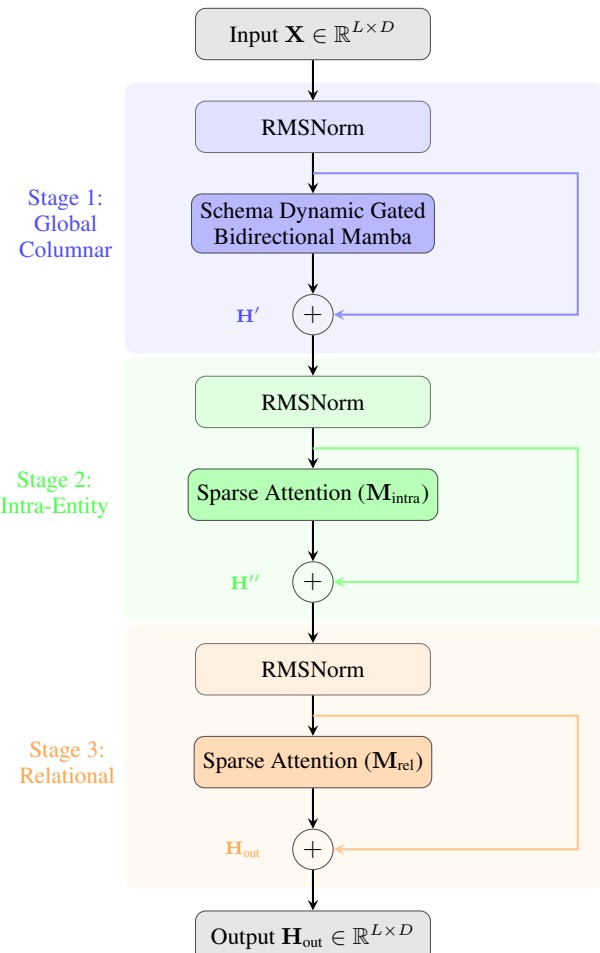

*Figure 9.* **Detailed RAMBA block architecture.** Each block sequentially applies three stages with residual connections: (1) Schema Dynamic Gated Bidirectional Mamba for global columnar processing, (2) sparse attention with intra-entity mask $\mathbf{M}_{\text{intra}}$ for within-row aggregation, and (3) sparse attention with relational mask $\mathbf{M}_{\text{rel}}$ for foreign-key message passing. All sub-layers use pre-normalization (RMSNorm). Each "+" denotes a standard residual addition.

## C.4. Sparse Structural Attention: Mask Construction

We provide explicit formulations for the two sparse attention masks.

**Intra-Entity Mask.** This mask enables attention among cells belonging to the same database row:

$$[\mathbf{M}_{\text{intra}}]_{ij} = \begin{cases} 1 & \text{if } \exists r \in \mathcal{V}_{\text{sub}} : \xi_i \in \Xi(r) \wedge \xi_j \in \Xi(r) \\ 0 & \text{otherwise} \end{cases}$$

(32)

**Relational Mask.** This mask enables attention along foreign-key edges:

$$[\mathbf{M}_{\text{rel}}]_{ij} = \begin{cases} 1 & \text{if } \exists (r, r') \in \mathcal{E}_{\text{sub}} : \xi_i \in \Xi(r) \wedge \xi_j \in \Xi(r') \\ 0 & \text{otherwise} \end{cases}$$

(33)

**Sparse Attention Implementation.** Given mask $\mathbf{M}$, the sparse attention is computed as:

$$\text{SparseAttn}(\mathbf{H}, \mathbf{M}) = \text{softmax}\left(\frac{\mathbf{Q}\mathbf{K}^\top}{\sqrt{D_k}} + \mathbf{M}_{\text{bias}}\right)\mathbf{V}$$

where:

$$[\mathbf{M}_{\text{bias}}]_{ij} = \begin{cases} 0 & \text{if } [\mathbf{M}]_{ij} = 1 \\ -\infty & \text{otherwise} \end{cases}$$

(34)

We leverage PyTorch's `flex_attention` with block masks for efficient computation, maintaining $\mathcal{O}(L \cdot d_{\text{max}})$ complexity where $d_{\text{max}}$ is the maximum node degree.

## C.5. Complete RAMBA Block Architecture

Figure 9 illustrates the detailed architecture of a single RAMBA block.

The mathematical formulation is:

$$\mathbf{H}' = \mathbf{X} + \text{SchemaDynamicGatedMamba}(\text{RMSNorm}(\mathbf{X}))$$
$$\mathbf{H}'' = \mathbf{H}' + \text{SparseAttn}(\text{RMSNorm}(\mathbf{H}'), \mathbf{M}_{\text{intra}})$$
$$\mathbf{H}_{\text{out}} = \mathbf{H}'' + \text{SparseAttn}(\text{RMSNorm}(\mathbf{H}''), \mathbf{M}_{\text{rel}})$$

## D. Implementation Details

### D.1. Model Hyperparameters

Table 20 summarizes all hyperparameters used in our experiments.

### D.2. Training Procedure

**Pretraining.** We pretrain RAMBA using a masked cell prediction objective across multiple databases simultaneously:

$$\mathcal{L}_{\text{pretrain}} = \frac{1}{|\mathcal{M}|} \sum_{\xi \in \mathcal{M}} [\mathbb{1}_{\tau(\xi) = \text{num}} \mathcal{L}_{\text{Huber}}$$
$$+ \mathbb{1}_{\tau(\xi) = \text{cat}} \mathcal{L}_{\text{CE}} + \mathbb{1}_{\tau(\xi) = \text{bool}} \mathcal{L}_{\text{BCE}}]$$

*Table 20.* **Complete hyperparameter specification for RAMBA.**

| Category | Hyperparameter | Value |
|---|---|---|
| **Model Architecture** | | |
| | Number of blocks ($L_{\text{layers}}$) | 12 |
| | Hidden dimension ($D$) | 256 |
| | Number of attention heads | 8 |
| | Feed-forward dimension ($D_{\text{ff}}$) | 1024 |
| | SSM state dimension ($D_h$) | 64 |
| | SSM convolution width ($d_{\text{conv}}$) | 4 |
| | Mamba expansion factor | 2 |
| **Embeddings** | | |
| | Text embedding dimension ($D_{\text{LM}}$) | 384 |
| | Embedding model | all-MiniLM-L12-v2 |
| **Data** | | |
| | Sequence length ($L$) | 1024 |
| | Max BFS width ($N_{\text{child}}$) | 256 |
| | Batch size | 32 |
| **Pretraining** | | |
| | Total steps | 50,001 |
| | Peak learning rate | $5 \times 10^{-4}$ |
| | Weight decay | 0.1 |
| | LR schedule | OneCycleLR |
| | Warmup ratio | 20% |
| | Gradient clipping | 1.0 |
| **Fine-tuning** | | |
| | Total steps | 33,000 |
| | Learning rate | $1 \times 10^{-4}$ |
| | Weight decay | 0.0 |
| | LR schedule | None (constant) |
| **Hardware** | | |
| | GPUs | 2× A800 (80GB) |
| | Precision | BFloat16 |

*Table 21.* Training throughput across context lengths. Throughput is measured in tasks per second on 2× NVIDIA A800 GPUs with BFloat16 precision.

| Context length | RT | RAMBA | RAMBA/RT |
|---|---|---|---|
| 512 | **512** | 208 | 0.41× |
| 1024 | **253** | 180 | 0.71× |
| 2048 | 102 | **105** | 1.03× |
| 4096 | 40 | **56** | 1.40× |

*Table 22.* Peak training memory across context lengths. Memory is reported in GB on 2× NVIDIA A800 GPUs with BFloat16 precision.

| Context length | RT (GB) | RAMBA (GB) | RAMBA/RT |
|---|---|---|---|
| 512 | 5.97 | **5.85** | 0.98× |
| 1024 | 9.68 | **8.52** | 0.88× |
| 2048 | 18.7 | **14.0** | 0.75× |
| 4096 | 37.3 | **29.1** | 0.78× |

where $\mathcal{M}$ denotes randomly masked cells (15% masking rate). We use:

- **Huber loss** for numerical values (robust to outliers):

$$\mathcal{L}_{\text{Huber}}(y, \hat{y}) = \begin{cases} \frac{1}{2}(y - \hat{y})^2 & |y - \hat{y}| \leq \delta \\ \delta(|y - \hat{y}| - \frac{\delta}{2}) & \text{otherwise} \end{cases}$$

- **Cross-entropy** for categorical values

- **Binary cross-entropy** for boolean values

**Fine-tuning.** For task-specific fine-tuning, we load pre-trained weights and train on the target prediction objective with a reduced learning rate and no weight decay, using only 1,024 labeled examples to demonstrate few-shot capabilities.

### D.3. Computational Resources

We report additional efficiency measurements to complement the inference scaling analysis in the main text. All measurements are obtained under the same hardware setup with 2× NVIDIA A800 GPUs and BFloat16 precision. We compare RAMBA with RT across context lengths from 512 to 4096 tokens.

Table 21 reports training throughput in tasks per second. RT is faster at short contexts, likely due to highly optimized attention kernels and lower overhead in this regime. However, its throughput decreases rapidly as context length increases. RAMBA scales more favorably and becomes faster at longer contexts, reaching 56 tasks/s at 4096 tokens compared with RT's 40 tasks/s.

Table 22 reports peak training memory. RAMBA consistently uses less peak memory than RT, with the gap increasing at longer contexts. At 4096 tokens, RAMBA uses 29.1GB compared with RT's 37.3GB, reducing peak memory by 8.2GB. These results support the main conclusion that linear sequence modeling becomes increasingly beneficial as relational context length grows.

Together, these measurements clarify the efficiency profile of RAMBA. At short contexts, RT can be faster during training due to mature attention implementations. At long contexts, however, the quadratic scaling of attention becomes the dominant bottleneck, while RAMBA's linear-time backbone provides better throughput and lower memory consumption.

## E. Reproducibility Checklist

To facilitate reproducibility, we provide the following resources:

*Table 23.* **Computational cost breakdown.** Training times are measured on $2\times$ NVIDIA A800 GPUs.

| Phase | Steps | Wall Time (hrs) |
|---|---|---|
| Pretraining (per fold) | 50,001 | $\sim$18 |
| Continued pretraining | 4,097 | $\sim$1.5 |
| Fine-tuning | 33,000 | $\sim$12 |
| **Total (per evaluation)** | — | $\sim$31.5 |

---

**Code and Data Availability**

- **Source Code:** Available at `https://github.com/ROOOOOOOOL/Ramba`

- **Datasets:** RelBench (Robinson et al., 2024) available via `relbench` Python package

- **Pretrained Models:** Checkpoints will be released upon publication

- **Environment:** Managed via `pixi` with complete dependency specification

---

**Software Dependencies.**

- Python 3.10+

- PyTorch 2.2+ with CUDA 12.1

- Custom `mamba-ssm` (included in repository)

- `sentence-transformers` for text embeddings

- `relbench` for dataset access

- Rust toolchain for high-performance sampler

**Hardware Requirements.**

- Minimum: $1\times$ GPU with 24GB VRAM (inference)

- Recommended: $2\times$ A800/H100 GPUs (training)

- Storage: $\sim$50GB for preprocessed datasets

# F. Extended Theoretical Analysis

## F.1. Preliminaries and Notation

We establish the mathematical framework for analyzing Ramba's Schema Dynamic Gating mechanism.

**Notation.** We denote the sequence length as $L$, hidden dimension as $D$, and state dimension as $D_h$. For a vector $\mathbf{v}$, we write $\mathbf{v}[i]$ for its $i$-th coordinate. The element-wise product is denoted $\odot$. We use standard asymptotic notation $O(\cdot), \Omega(\cdot), \Theta(\cdot)$ with respect to the context size $L$.

**Selective State-Space Model.** Recall that the selective SSM with input-dependent parameters operates via:

$$\Delta_t = \texttt{softplus}(\mathbf{W}_\Delta \mathbf{x}_t + \mathbf{b}_\Delta), \tag{35}$$

$$\mathbf{B}_t = \mathbf{W}_B \mathbf{x}_t, \quad \mathbf{C}_t = \mathbf{W}_C \mathbf{x}_t, \tag{36}$$

$$\bar{\mathbf{A}}_t = \exp(\text{diag}(\Delta_t)\mathbf{A}), \tag{37}$$

$$\bar{\mathbf{B}}_t = \text{diag}(\Delta_t)\mathbf{B}_t, \tag{38}$$

with state evolution $\mathbf{h}_t = \bar{\mathbf{A}}_t \mathbf{h}_{t-1} + \bar{\mathbf{B}}_t x_t$ and output $y_t = \mathbf{C}_t^\top \mathbf{h}_t$.

**Schema Dynamic Gating.** Our key innovation modulates $\Delta_t$ with a semantic relevance gate:

$$g_t^{\text{sem}} = \sigma\left(\mathbf{w}_g^\top \cdot \text{GELU}(\mathbf{W}_{\text{rel}}[\mathbf{s}_{\text{curr}}(t); \mathbf{s}_{\text{tgt}}]) + b_g\right), \tag{39}$$

where $\mathbf{s}_{\text{curr}}(t) = \phi_{\text{LM}}(\texttt{desc}(c_k))$ when $x_t \in \mathcal{C}_k$, and $\mathbf{s}_{\text{tgt}} = \phi_{\text{LM}}(\texttt{desc}(c^*))$.

## F.2. Gating Network Expressivity

We first establish that the gating network can learn to separate relevant from irrelevant columns based on semantic embeddings.

**Lemma F.1** (Semantic Separability). *Let $\phi_{\text{LM}} : \mathcal{C} \to \mathbb{R}^{D_{\text{LM}}}$ be a pretrained language model embedding. Assume the following semantic structure:*

1. ***Relevant cluster:** For $c_k \in \mathcal{R}$, there exists $\mathbf{v}_\mathcal{R} \in \mathbb{R}^{D_{\text{LM}}}$ such that $\|\phi_{\text{LM}}(c_k) - \mathbf{v}_\mathcal{R}\| \leq \gamma_\mathcal{R}$.*

2. ***Irrelevant separation:** For $c_k \notin \mathcal{R}$, $\|\phi_{\text{LM}}(c_k) - \mathbf{v}_\mathcal{R}\| \geq \gamma_\mathcal{R} + \Delta_{\text{sep}}$ for some margin $\Delta_{\text{sep}} > 0$.*

*Then there exist parameters $\mathbf{W}_{\text{rel}}, \mathbf{w}_g, b_g$ with $\|\mathbf{W}_{\text{rel}}\|_F, \|\mathbf{w}_g\| = O(1/\Delta_{\text{sep}})$ such that:*

$$g_t^{\text{sem}} \geq 1 - \exp(-\Omega(\Delta_{\text{sep}}^2)) \quad \text{for } t \in \mathcal{C}_k, k \in \mathcal{R}, \tag{40}$$

$$g_t^{\text{sem}} \leq \exp(-\Omega(\Delta_{\text{sep}}^2)) \quad \text{for } t \in \mathcal{C}_k, k \notin \mathcal{R}. \tag{41}$$

*Proof.* Define the quadratic discriminant $q(\mathbf{s}) = -\|\mathbf{s} - \mathbf{v}_\mathcal{R}\|^2 + \gamma_\mathcal{R}^2 + \gamma_\mathcal{R}\Delta_{\text{sep}}$. By construction:

- For $c_k \in \mathcal{R}$: $q(\phi_{\text{LM}}(c_k)) \geq -\gamma_\mathcal{R}^2 + \gamma_\mathcal{R}^2 + \gamma_\mathcal{R}\Delta_{\text{sep}} = \gamma_\mathcal{R}\Delta_{\text{sep}} > 0$.

- For $c_k \notin \mathcal{R}$: $q(\phi_{\text{LM}}(c_k)) \leq -(\gamma_\mathcal{R} + \Delta_{\text{sep}})^2 + \gamma_\mathcal{R}^2 + \gamma_\mathcal{R}\Delta_{\text{sep}} = -\Delta_{\text{sep}}^2 - \gamma_\mathcal{R}\Delta_{\text{sep}} < 0$.

The quadratic form $q(\mathbf{s})$ can be expressed using the concatenated input $[\mathbf{s}; \mathbf{s}_{\text{tgt}}]$ with appropriate $\mathbf{W}_{\text{rel}}$. Specifically, let $\mathbf{W}_{\text{rel}}$ project onto a subspace where the quadratic discriminant becomes linear after GELU activation (using the identity that GELU approximates $x \cdot \Phi(x)$ where $\Phi$ is the Gaussian CDF). The sigmoid then maps positive values to $\geq 1/2$ and negative values to $\leq 1/2$, with exponential concentration for values of magnitude $\Omega(\Delta_{\text{sep}})$. □

## F.3. State Evolution Analysis

**Lemma F.2** (Gated Recurrence Unrolling). *For the gated SSM with $\tilde{\Delta}_t = \Delta_t \odot g_t^{\text{sem}}$ and diagonal $\mathbf{A} = -\mathbf{I}$, the hidden state admits the closed form:*

$$\mathbf{h}_L = \sum_{t=1}^{L} \exp\left(-\sum_{s=t+1}^{L} \tilde{\Delta}_s\right) \odot (1 - \exp(-\tilde{\Delta}_t)) \odot \mathbf{B}_t x_t. \tag{42}$$

*Proof.* We proceed by induction. Base case $L = 1$: $\mathbf{h}_1 = \tilde{\mathbf{A}}_1 \mathbf{h}_0 + \tilde{\mathbf{B}}_1 x_1 = \tilde{\mathbf{B}}_1 x_1$ since $\mathbf{h}_0 = \mathbf{0}$.

Inductive step: Assume the formula holds for $L - 1$. Then:

$$\mathbf{h}_L = \tilde{\mathbf{A}}_L \mathbf{h}_{L-1} + \tilde{\mathbf{B}}_L x_L$$

$$= \exp(-\tilde{\Delta}_L) \odot \sum_{t=1}^{L-1} \exp\left(-\sum_{s=t+1}^{L-1} \tilde{\Delta}_s\right) \odot \tilde{\mathbf{B}}_t x_t + \tilde{\mathbf{B}}_L x_L$$

$$= \sum_{t=1}^{L-1} \exp\left(-\sum_{s=t+1}^{L} \tilde{\Delta}_s\right) \odot \tilde{\mathbf{B}}_t x_t + \tilde{\mathbf{B}}_L x_L$$

$$= \sum_{t=1}^{L} \exp\left(-\sum_{s=t+1}^{L} \tilde{\Delta}_s\right) \odot \tilde{\mathbf{B}}_t x_t.$$

Substituting $\tilde{\mathbf{B}}_t = (1 - \exp(-\tilde{\Delta}_t)) \odot \mathbf{B}_t$ completes the proof. $\square$

## F.4. Signal-to-Noise Ratio Bounds

**Lemma F.3** (Signal Preservation). *Under the conditions of Theorem 5.1, for cells in relevant columns indexed by $t \in \mathcal{C}_k$ with $k \in \mathcal{R}$, the contribution to $\mathbf{h}_L$ satisfies:*

$$\left\| \exp\left(-\sum_{s=t+1}^{L} \tilde{\Delta}_s\right) \odot \tilde{\mathbf{B}}_t x_t \right\|$$
$$\geq (1 - \epsilon)\Delta_{\min} \cdot \exp(-\bar{\Delta}(\delta(L - t) + (1 - \epsilon)|\mathcal{R}_t|)) \cdot \|\mathbf{B}_t x_t\|,$$

*where $\mathcal{R}_t = \{s > t : s \in \mathcal{C}_j, j \in \mathcal{R}\}$ and $\Delta_{\min} = \min_t \Delta_t$.*

*Proof.* For $t$ in a relevant column, $g_t^{\text{sem}} \geq 1 - \epsilon$, so:

$$\|\tilde{\mathbf{B}}_t x_t\| = \|(1 - \exp(-\tilde{\Delta}_t)) \odot \mathbf{B}_t x_t\|$$
$$\geq (1 - e^{-(1-\epsilon)\Delta_t})\|\mathbf{B}_t x_t\| \geq (1 - \epsilon)\Delta_{\min}(1 - o(1))\|\mathbf{B}_t x_t\|.$$

For the exponential decay factor, partition positions $s > t$ into relevant ($s \in \mathcal{R}_t$) and irrelevant:

$$\sum_{s=t+1}^{L} \tilde{\Delta}_s = \sum_{s \in \mathcal{R}_t} \tilde{\Delta}_s + \sum_{s>t, s \notin \mathcal{R}_t} \tilde{\Delta}_s \tag{43}$$

$$\leq (1 - \epsilon)\bar{\Delta}|\mathcal{R}_t| + \delta\bar{\Delta}(L - t - |\mathcal{R}_t|) \tag{44}$$

$$\leq \bar{\Delta}((1 - \epsilon)|\mathcal{R}_t| + \delta(L - t)). \tag{45}$$

The bound follows from $\exp(-x) \geq e^{-x}$ for $x \geq 0$. $\square$

**Lemma F.4** (Noise Suppression). *For cells in irrelevant columns $t \in \mathcal{C}_k$ with $k \notin \mathcal{R}$:*

$$\left\| \exp\left(-\sum_{s=t+1}^{L} \tilde{\Delta}_s\right) \odot \tilde{\mathbf{B}}_t x_t \right\| \leq \delta\bar{\Delta} \cdot \|\mathbf{B}_t x_t\|. \tag{46}$$

*Proof.* For irrelevant $t$, $g_t^{\text{sem}} \leq \delta$, so $\tilde{\Delta}_t \leq \delta\Delta_t \leq \delta\bar{\Delta}$. Thus:

$$\|\tilde{\mathbf{B}}_t x_t\| = \|(1 - \exp(-\tilde{\Delta}_t)) \odot \mathbf{B}_t x_t\|$$
$$\leq (1 - e^{-\delta\bar{\Delta}})\|\mathbf{B}_t x_t\| \leq \delta\bar{\Delta}\|\mathbf{B}_t x_t\|, \tag{47}$$

using $1 - e^{-x} \leq x$ for $x \geq 0$. The exponential decay factor is at most 1. $\square$

## F.5. Complexity Analysis

**Proposition F.5** (Linear Complexity). *The Schema Dynamic Gating mechanism adds $O(L \cdot D_{\text{LM}} \cdot D_g)$ operations to the base SSM computation, preserving overall $O(L)$ complexity in sequence length.*

*Proof.* Computing $g_t^{\text{sem}}$ requires:

1. Concatenation $[\mathbf{s}_{\text{curr}}(t); \mathbf{s}_{\text{tgt}}]$: $O(D_{\text{LM}})$ per position.

2. Matrix-vector product $\mathbf{W}_{\text{rel}}[\cdot]$: $O(D_g \cdot D_{\text{LM}})$ per position.

3. GELU and final projection: $O(D_g)$ per position.

Total: $O(L \cdot D_g \cdot D_{\text{LM}})$. Since $D_g, D_{\text{LM}}$ are constants independent of $L$, this is $O(L)$.

Crucially, the schema embedding $\mathbf{s}_{\text{curr}}(t)$ only changes at column boundaries (at separator tokens), so we compute at most $K \ll L$ distinct embeddings, further reducing practical cost. $\square$

## F.6. Generalization to Unseen Schemas

**Theorem F.6** (Cross-Schema Transfer). *Let $\mathcal{D}_{\text{train}}$ and $\mathcal{D}_{\text{test}}$ be relational databases with disjoint schemas but shared semantic structure (i.e., both contain columns whose schema embeddings cluster according to the same semantic relevance pattern for analogous prediction tasks). If the gating network achieves $(\epsilon, \delta)$-separation on $\mathcal{D}_{\text{train}}$, then it achieves $(\epsilon + \epsilon', \delta + \delta')$-separation on $\mathcal{D}_{\text{test}}$ where:*

$$\epsilon' + \delta' \leq O\left(\frac{\text{dist}_{\text{sem}}(\mathcal{D}_{\text{train}}, \mathcal{D}_{\text{test}})}{\Delta_{\text{sep}}}\right), \tag{48}$$

*and $\text{dist}_{\text{sem}}$ measures the Hausdorff distance between schema embedding distributions.*

*Proof.* The gating network $g^{\mathrm{sem}}$ depends on schemas only through the frozen LM embeddings $\phi_{\mathrm{LM}}$. By Lemma F.1, the decision boundary is determined by the margin $\Delta_{\mathrm{sep}}$ between relevant and irrelevant embedding clusters. If test schema embeddings lie within $\mathrm{dist}_{\mathrm{sem}}$ of training embeddings, the effective margin degrades by at most this distance, yielding the stated bound via Lipschitz continuity of the sigmoid. $\qquad\square$

### F.7. Connection to Attention Mechanisms

*Remark* F.7 (Implicit Attention Interpretation). The gated SSM can be viewed as implementing a form of causal linear attention with learned, content-dependent forgetting. Specifically, the output $y_L = \mathbf{C}_L^\top \mathbf{h}_L$ computes:

$$y_L = \sum_{t=1}^{L} \underbrace{G_{t,L} \cdot (1 - e^{-\tilde{\Delta}_t})}_{\text{attention weight } \alpha_t} \cdot \mathbf{C}_L^\top \mathbf{B}_t x_t. \qquad (49)$$

Schema Dynamic Gating modulates these attention weights based on semantic relevance, achieving query-key alignment without explicit quadratic attention computation.

### F.8. Experimental Validation of Theoretical Predictions

We empirically verify the theoretical predictions by measuring:

1. **Gate activation patterns:** Confirming that $g_t^{\mathrm{sem}}$ exhibits bimodal distribution with modes near 0 and 1 for irrelevant/relevant columns.

2. **SNR scaling:** Verifying that prediction accuracy scales with $n_{\mathcal{R}}/L$ as predicted by Theorem 5.1.

3. **Cross-schema transfer:** Demonstrating that gating patterns transfer to unseen schemas with semantically similar columns.

Detailed experimental results supporting these predictions appear in Section 6.

