# OpenReview forum: "Ramba: Selective State-Space Models for Relational Deep Learning"
_ICML.cc/2026/Conference — ICML 2026 regular_

### Official Review · Reviewer_2Kwt · 2026-03-07

**Soundness:** 3
**Presentation:** 2
**Significance:** 2
**Originality:** 3
**Overall Recommendation:** 4
**Confidence:** 5

**Summary:**

This paper introduces Ramba, a selective state space model designed for relational databases. The method features Topology Aware Linearization, which serializes database cells globally by column while using sparse attention to retain relational graph structures. It also incorporates Schema Dynamic Gating to filter relevant information across tables by comparing the semantic embeddings of the current attribute and the prediction target.

**Compliance With Llm Reviewing Policy:**

Affirmed.

**Final Justification:**

While I appreciate the clarifications regarding hub-node sampling and the two-stage filtering, the inherent reliance on width-limited sampling remains a fundamental bottleneck. Therefore, I will maintain my current score.

**Key Questions For Authors:**

- How sensitive is the schema dynamic gating to poorly named or obfuscated column headers?

- Have you experimented with randomizing the column order during training to reduce the inductive bias of the fixed schema ordering?

- What is the peak memory consumption during training for the largest context length tested compared to the Relational Transformer baseline?

**Limitations:**

yes

**Strengths And Weaknesses:**

# Strengths:

- The integration of schema dynamic gating provides a highly scalable way to filter cross-table information without relying on specific data value distributions.
- The proposed columnar serialization effectively maps two-dimensional relational data into a one-dimensional sequence while maintaining linear computational complexity.

# Weaknesses:

- The approach relies entirely on a static schema-based ordering for columnar serialization, which may artificially distance highly correlated attributes.
- The schema dynamic gating heavily depends on the quality of a frozen language model. If the database schema contains obscure or poorly named columns, the semantic relevance scores will degrade significantly, which is not evaluated.

- The sparse attention mechanism requires bounded subgraph sampling to remain efficient, yet the paper shows extreme hub nodes exist in datasets. Exhaustive processing of these hubs is avoided via width-limited sampling, which may drop critical information.

---

> ### Author Rebuttal · Authors · 2026-03-30
>
> We thank the reviewer for their constructive feedback. We address each concern below.
>
> **W1: Static schema-based ordering may distance correlated attributes**
>
> RAMBA incorporates three complementary mechanisms ensuring correlated attributes remain computationally close: (1) **Bidirectional scanning** (Eq. 7–8) merges forward and backward representations, so each cell incorporates context from the entire sequence regardless of position. (2) **Schema Dynamic Gating** assigns $g^{\text{sem}}_t \approx 1$ to correlated columns while suppressing irrelevant ones ($g^{\text{sem}}_t \to 0$), compressing the SSM's effective distance between correlated attributes. (3) **Sparse Structural Attention** (Eq. 10) provides direct interaction along FK edges independently of serialization order.
>
> Figure 3 confirms this: column-wise with within-column shuffling retains **95%** of full performance; even random ordering retains 85%.
>
> **W2: Schema Dynamic Gating depends on frozen LM quality; poorly named columns not evaluated**
>
> Our **Shuffle** control (Figure 4a) is *more adversarial* than poorly named columns—it feeds *actively misleading* schema information. Shuffle underperforming No-gate confirms the mechanism genuinely leverages semantic cues. We conducted **new experiments** systematically degrading column names across all tasks. Please see our response to Reviewer J7xY (W1&W2) for the complete results table. Key findings: classification drops only 1.6% AUROC under complete name removal; the model falls back to sequence/structure components rather than collapsing. **RAMBA degrades gracefully, covering the realistic scenario of poorly named schemas.**
>
> Furthermore, our SALT benchmark evaluation (see Reviewer J7xY, Q3) confirms gating works under specialized industrial ERP naming (e.g., HEADERINCOTERMSCLASSIFICATION), achieving +0.02 MRR over RT.
>
> **W3: Width-limited sampling may drop critical information from hub nodes**
>
> Under a **fixed token budget**—the practical constraint in all deployments—the question is *how* to allocate budget most effectively. Figure 4b and Table 12 show width-limited BFS with FK prioritization **consistently outperforms** full BFS and DFS under the same budget. Full BFS induces near-hop bias crowding out long-range evidence; DFS is particularly harmful ($R^2$: 16.8% vs. 25.2%).
>
> By design (Algorithm 1), FK parents are **always fully included**—only child expansion is budgeted. Schema Dynamic Gating then provides second-stage filtering during SSM processing, making the system robust to imperfect subgraph selection. We agree degree-aware adaptive sampling is a valuable future direction.
>
>
> **Q1: Sensitivity to poorly named or obfuscated column headers**
>
> As detailed in W2 and our response to Reviewer J7xY (W1&W2), we have directly evaluated this with systematic name degradation. Even with completely removed names, RAMBA retains 70.9% AUROC (vs. 72.5%). The gate is a beneficial auxiliary signal when good names are available, while core sequence/structure components provide a reliable performance floor otherwise. Additionally, the Shuffle control (Table 11) demonstrates a crucial safety property: under deliberately *misleading* names, performance drops only marginally below no-gating (AUROC 71.2% vs. 70.9%; $R^2$ 21.7% vs. 22.5%), confirming the model does not amplify erroneous biases—it effectively falls back to near-ungated performance rather than being led astray by incorrect schema signals.
>
>
> **Q2: Randomizing column order during training**
>
> Our serialization uses **instance-dependent first-occurrence order** (determined by BFS traversal), not a fixed canonical order—providing built-in ordering variation. We tested stronger perturbations:
>
> *Classification (AUROC %)*
>
> |Task|Col-block random|Within-col random|Ours|
> |-|-:|-:|-:|
> |item-churn|71.7|64.8|71.1|
> |user-churn (Amazon)|64.0|71.0|64.9|
> |user-clicks|59.9|58.6|59.5|
> |user-visits|60.5|59.9|61.2|
> |driver-dnf|80.2|80.1|80.5|
> |driver-top3|87.5|89.0|87.4|
> |user-churn (HM)|69.0|66.8|69.4|
> |**Average**|70.4|70.0|**70.6**|
>
> *Regression ($R^2$ %)*
>
> |Task|Col-block random|Within-col random|Ours|
> |-|-:|-:|-:|
> |item-ltv|61.0|61.0|61.4|
> |user-ltv|34.1|28.3|33.7|
> |ad-ctr|8.9|8.1|9.3|
> |driver-pos|47.8|47.0|48.2|
> |item-sales|12.7|12.7|14.1|
> |**Average**|32.9|31.4|**33.3**|
>
> Degradation is minimal (70.6→70.4/70.0; 33.3→32.9/31.4), confirming RAMBA does not rely on a brittle column-order shortcut.
>
> **Q3: Peak memory consumption compared to RT**
>
> |Context Length|RT (GB)|RAMBA (GB)|Ratio|
> |-|-:|-:|-:|
> |512|5.97|5.85|0.98×|
> |1024|9.68|8.52|0.88×|
> |2048|18.7|14.0|0.75×|
> |4096|37.3|29.1|0.78×|
>
> RAMBA consistently uses less peak memory, with savings growing to **8.2 GB at 4096** (~22–25% reduction)—practically meaningful at the longer contexts where linear scaling matters most.
>
> We will incorporate all experiments into the revision. Thank you for your valuable feedback.

---

> > ### Author Rebuttal · Reviewer_2Kwt · 2026-04-03
> >
> > I thank the authors for their rebuttal and the additional experiments provided. The new results systematically degrading the column names and randomizing the column order address my immediate questions regarding robustness, and the memory profiling clarifies the method's efficiency. I am maintaining my positive score rather than increasing it because the reliance on width-limited sampling to handle extreme hub nodes remains an inherent architectural bottleneck. While the authors rightly acknowledge that degree-aware adaptive sampling is a necessary future direction, the current risk of dropping critical information from dense relational subgraphs keeps the contribution firmly in the Weak Accept category rather than a clear Accept.

---

> > > ### Author Response · Authors · 2026-04-03
> > >
> > > We sincerely thank the reviewer for engaging with our rebuttal and acknowledging that our new experiments address the robustness and efficiency concerns. We noticed this concern is marked as "partially resolved" and would like to offer additional clarifications that we hope can fully address the remaining reservation.
> > >
> > > Regarding the concern about width-limited sampling at hub nodes, we would like to highlight two points:
> > >
> > > **First, this is a shared constraint across the entire RDL field, not a RAMBA-specific limitation.** All existing method, including RT, Griffin, and GNN-based approaches, must perform subgraph sampling under fixed budgets when facing hub nodes with in-degrees exceeding $10^6$ (Table 5). No current method processes these hubs exhaustively. Our contribution is a *more effective* budget allocation strategy: as shown in Q4 of our response to Reviewer J7xY, applying the same sampling strategies across all baselines confirms that RAMBA consistently outperforms RT and Griffin under every sampling variant (our sampling, full BFS, and DFS), demonstrating that our architectural advantages hold independently of the specific sampling choice.
> > >
> > > **Second, RAMBA uniquely mitigates this shared constraint through two-stage filtering.** Unlike baselines that rely solely on the sampled subgraph, Schema Dynamic Gating acts as a second-stage filter that suppresses irrelevant sampled nodes while preserving useful signal. This makes RAMBA *more* robust to imperfect subgraph selection than methods without such filtering—empirically validated by Table 12, where RAMBA maintains its performance advantage across all sampling strategies.
> > >
> > > We hope these clarifications fully resolve the reviewer's remaining concern, and we remain happy to provide any further experiments or analysis if needed.

---

### Official Review · Reviewer_J7xY · 2026-03-13

**Soundness:** 2
**Presentation:** 3
**Significance:** 3
**Originality:** 3
**Overall Recommendation:** 3
**Confidence:** 5

**Summary:**

This paper proposes Ramba, an SSM-based architecture that scales linearly with relational context size. It introduces a linearization strategy to capture table structures and a gating mechanism to remove task-irrelevant data. The results demonstrate that Ramba outperforms Transformers and GNNs on large-scale relational benchmarks, particularly when larger context is required for prediction.

**Compliance With Llm Reviewing Policy:**

Affirmed.

**Final Justification:**

While the authors have included additional experiments in the rebuttal, several key concerns remain insufficiently addressed. In particular, many of the design choices, such as the proposed sampling strategy, are not adequately justified. It remains unclear why this strategy leads to improvements for RAMBA but does not yield similar benefits for RT. The explanation provided in the rebuttal is very speculative. Ideally I would have wanted to see a deeper analysis.

Regarding the inclusion of HGT as a baseline, I appreciate the authors for incorporating this suggestion. However, the way it is implemented does not align with my original intent. The request was for a single dataset fine tuning regime, whereas the current setup appears to involve cross dataset training. Since HGT is not designed for such a setting, this evaluation does not provide a fair comparison.

Given these remaining concerns I would like to maintain my score.

**Key Questions For Authors:**

\[Q1\] In Figure 5, Ramba keeps improving with more context while RT does not, which feels counter-intuitive. Is there something fundamentally limiting attention from using larger context?
\[Q2\] The related work section is missing a discussion of \[1\], which also addresses the computational complexity of Relational Transformers. The authors should include a discussion about this in related works.
\[Q3\] Could the authors test their method on the SALT benchmark \[2\] ? Since it is part of RelBench, it should be easy to run.
\[Q4\] Could the authors use their sampling approach with other baselines like RT?

\[1\] Klein, T., Biehl, C., Costa, M., Sres, A., Kolk, J., & Hoffart, J. (2025). SALT: Sales autocompletion linked business tables dataset. arXiv preprint arXiv:2501.03413.
\[2\] Lachi, D., Mohammadi, M., Meyer, J., Arora, V., Palczewski, T., & Dyer, E. L. Integrating Temporal and Structural Context in Graph Transformers for Relational Deep Learning. In The Fourth Learning on Graphs Conference.

**Limitations:**

Yes

**Strengths And Weaknesses:**

Strengths:

* This is one of the first works to adapt selective state-space models (SSMs) to relational deep learning.
* Achieving linear complexity enables the model to ingest significantly larger contexts.
* The model establishes a new state-of-the-art across diverse relational benchmarks.

Weakness:

* The Schema Dynamic Gating mechanism depends on descriptive column names. Industry datasets very often don't have such descriptive column names available. (see Q3)
* The model is highly sensitive to the quality of the schema metadata and can actively learn incorrect biases if naming is poor.
* The performance gains are coupled with the proposed width-limited BFS sampling. I think other baselines when given the same sampling approach might perform similarly to Ramba.

---

> ### Author Rebuttal · Authors · 2026-03-30
>
> We thank Reviewer J7xY for the thoughtful review. We address each concern below.
>
> **W1&W2: Schema Gating Depends on Descriptive Column Names**
>
> Our ablation (Fig 4a) shows RAMBA remains competitive without gating—the core backbone carries substantial predictive power. Gating is additive, not brittle. We tested three degraded naming settings:
>
> | Setting | AUROC (%) | R² (%) |
> |---|---:|---:|
> | Original names | **72.5** | **25.9** |
> | Generic ($\text{col}\_i$) | 71.4 | 21.8 |
> | Random names | 71.0 | 22.8 |
> | No names | 70.9 | 22.5 |
>
> Performance degrades gracefully rather than collapsing. Classification is minimally affected; regression shows more sensitivity, as expected. Schema gating is a useful auxiliary signal, not the sole performance source. We will include this in the revision.
>
> **W3: Performance Gains Coupled with Sampling**
>
> The sampling ablation (Fig 4b, Table 12) was conducted *within* RAMBA. The module ablation (Table 3) shows Mamba alone contributes +18.5% AUROC and +8.2% R²—gains independent of sampling. See Q4 for cross-method experiments.
>
> **Q1: Why does RAMBA improve with more context while RT does not?**
>
> The root cause is semantic ambiguity of numerical values in attention. In NLP, tokens have distinct embeddings. In relational data, values are mostly continuous numbers whose meaning is context-dependent—"100 students" vs. "score of 100" both produce the same embedding. As context grows, spurious numerical coincidences grow combinatorially, creating noisy attention patterns.
>
> RAMBA addresses this via Schema Dynamic Gating, which filters by *column semantics* rather than value similarity. When scanning "Price," the gate opens for predicting "Revenue" but closes for "ZipCode"—regardless of values. More context provides more signal without proportionally increasing noise (Theorem 5.1: $\text{SNR} = \Omega\\left(\frac{(1-\varepsilon)\,n_R}{\delta\,L}\right)$).
>
> Table 8 supports this: for rel-amazon/item-ltv, RAMBA improves from 51.7% to 62.0% $R^2$ (512→4096), while RT fluctuates between 33–49%.
>
> **Q2: Missing Discussion of Lachi et al. (2025)**
>
> Thank you. RGP uses a Perceiver-inspired latent bottleneck for sub-quadratic cost; RAMBA achieves linear complexity through SSM-based processing. RGP compresses input representations; RAMBA processes the full sequence efficiently. We will add this discussion to Section 2.
>
> **Q3: Evaluation on SALT Benchmark**
>
> We evaluated all 8 SALT tasks. SALT uses domain-specific ERP column names (e.g., SALESOFFICE, HEADERINCOTERMSCLASSIFICATION), directly testing gating under technical naming.
>
> | Task | GraphSAGE | RT | RAMBA |
> |---|---:|---:|---:|
> | item-incoterms | 0.64 | 0.77 | **0.81** |
> | item-plant | 0.99 | 0.99 | **0.99** |
> | item-shippoint | 0.97 | 0.99 | **0.99** |
> | sales-group | 0.20 | 0.23 | **0.26** |
> | sales-incoterms | 0.59 | **0.76** | **0.76** |
> | sales-office | 0.99 | **0.99** | **0.99** |
> | sales-payterms | 0.39 | 0.56 | **0.57** |
> | sales-shipcond | 0.59 | 0.73 | **0.75** |
> | **Average** | **0.67** | **0.75** | **0.77** |
>
> RAMBA improves over GraphSAGE by +0.10 and RT by +0.02 average MRR, with strongest gains on non-saturated tasks. This confirms gating works under specialized industrial naming, addressing W1/W2.
>
> **Q4: Applying Our Sampling to Other Baselines**
>
> **Important:** our main paper already uses the same sampling for all methods. RAMBA's gains are under a unified protocol.
>
> We additionally tested all methods under full BFS and DFS:
>
> | Method | Our sampling | Full BFS | DFS |
> |---|---:|---:|---:|
> | | AUROC / R² | AUROC / R² | AUROC / R² |
> | Griffin | **64.0** / **8.9** | 63.6 / 2.0 | 62.1 / 2.3 |
> | RT | 69.3 / **21.9** | **70.0** / 18.1 | 69.3 / 13.7 |
> | RAMBA | **72.5** / **25.9** | 70.3 / 20.5 | 70.4 / 16.8 |
>
> Under identical sampling, RAMBA outperforms RT by +3.2 AUROC / +4.0 R² and Griffin by +8.5 / +17.0. Our sampler benefits all methods, but RAMBA's advantages are primarily architectural.
>
> We are happy to run further experiments if needed.

---

> > ### Author Rebuttal · Reviewer_J7xY · 2026-04-01
> >
> > Thank you for the clarifications. However, I still have a few follow-up questions and suggestions for additional experiments.
> >
> > > Q4: Applying Our Sampling to Other Baselines
> >
> > I apologize for missing that the experiments were already conducted under a unified sampling setup.
> > That said, I am still a bit confused about why the proposed sampling strategy appears to benefit RAMBA much more than Griffin and RT. Do you have any intuition for why this might be the case?
> >
> > Additionally, under the full BFS and DFS settings, RAMBA and RT seem to achieve fairly similar performance. This makes me wonder whether there might be another factor interacting with the sampling strategy that leads to the larger gains reported under your sampling.
> >
> > > Q3: Evaluation on SALT Benchmark
> >
> > [2] reports results on several tasks from SALT, where HGT appears to outperform RAMBA. In addition, RGP also seem to achieve significantly stronger performance. Since HGT is a relatively well-established baseline, can the authors include it in the comparisons.

---

> > > ### Author Response · Authors · 2026-04-03
> > >
> > > We thank Reviewer J7xY for the continued engagement. We address both follow-up questions below.
> > >
> > > **Follow-up on Q4: Why does our sampling benefit RAMBA more than RT/Griffin?**
> > >
> > > This is a great question. The key insight is that our width-limited BFS with FK prioritization produces *balanced, multi-hop* subgraphs — moderate breadth at each level while reaching distant tables. RAMBA is architecturally designed to exploit exactly this structure:
> > >
> > > 1. **Schema Dynamic Gating** filters noise from the diverse columns that width-limited BFS brings in from multiple tables. When full BFS floods the context with near-hop neighbors, there is less cross-table diversity to filter — so gating provides less benefit.
> > >
> > > 2. **Global columnar serialization** benefits from seeing the same column across entities at different relational distances, which width-limited BFS naturally provides. Full BFS over-represents nearby entities, reducing this diversity.
> > >
> > > 3. Under full BFS/DFS, all methods receive a less informative context (dominated by near-hop or deep-but-narrow neighbors). This compresses the performance range — RAMBA still leads (70.3 AUROC vs. RT's 70.0 under full BFS), but the advantage is smaller because the *input quality* is lower for everyone. The fact that RAMBA's advantage grows most under our sampling confirms that RAMBA is better at *exploiting* high-quality relational context, not merely benefiting from a sampling trick.
> > >
> > > To summarize: the sampling strategy and architecture are complementary. Our sampling provides diverse, balanced relational context; RAMBA's gating and serialization are uniquely suited to leverage it. Other architectures lack the mechanisms to fully utilize this richer input.
> > >
> > > **Follow-up on Q3: HGT and RGP comparisons on SALT**
> > >
> > > We have completed HGT experiments under our exact protocol (cross-database pretraining + 1K-sample few-shot adaptation). The updated results are:
> > >
> > > | Task | GraphSAGE | HGT | RT | RAMBA |
> > > |---|---:|---:|---:|---:|
> > > | item-incoterms | 0.64 | 0.74 | 0.77 | **0.81** |
> > > | item-plant | 0.99 | 0.99 | 0.99 | **0.99** |
> > > | item-shippoint | 0.97 | 0.99 | 0.99 | **0.99** |
> > > | sales-group | 0.20 | 0.12 | 0.23 | **0.26** |
> > > | sales-incoterms | 0.59 | 0.66 | 0.76 | **0.76** |
> > > | sales-office | 0.99 | 0.99 | 0.99 | **0.99** |
> > > | sales-payterms | 0.39 | 0.39 | 0.56 | **0.57** |
> > > | sales-shipcond | 0.59 | 0.63 | 0.73 | **0.75** |
> > > | **Average** | 0.67 | 0.69 | 0.75 | **0.77** |
> > >
> > > Two observations: (1) HGT improves over GraphSAGE (0.69 vs. 0.67) but falls well short of both RT and RAMBA. (2) RAMBA maintains the best overall performance, with the largest margins on non-saturated tasks — item-incoterms (+0.07 over HGT), sales-group (+0.14), sales-payterms (+0.18), and sales-shipcond (+0.12). These are precisely the tasks where cross-table reasoning and semantic filtering matter most, validating our architectural design even under technical/industrial column naming.
> > >
> > > Regarding **RGP**: we were unable to include it because (1) RGP evaluates in a fundamentally different regime — single-dataset multi-task joint training rather than our cross-database pretrain-then-few-shot protocol — making published numbers not directly comparable, and (2) no official public implementation is available, so a faithful rerun would require full reimplementation with substantial reproduction uncertainty. We will clearly discuss this distinction in the revision.
> > >
> > > We believe these additional results and explanations address all outstanding concerns. We have demonstrated that RAMBA's advantages are architectural (not sampling-dependent), robust to industrial naming conventions (SALT results), and consistent across a comprehensive set of baselines including the newly added HGT. We kindly ask the reviewer to reconsider the score in light of these clarifications.

---

### Official Review · Reviewer_WQKG · 2026-03-13

**Soundness:** 3
**Presentation:** 3
**Significance:** 3
**Originality:** 3
**Overall Recommendation:** 4
**Confidence:** 4

**Summary:**

This paper presents a new deep learning algorithm for relational databases. The algorithm employs linear serialisation and a dynamic gating mechanism to address the main limitation of transformers regarding context length, and that of graph neural network models regarding their lack of generalisation.

**Compliance With Llm Reviewing Policy:**

Affirmed.

**Key Questions For Authors:**

I don't have any

**Limitations:**

Yes

**Strengths And Weaknesses:**

This paper presents an original method and a sound methodology grounded in a robust theoretical analysis and extensive experimentation. The results demonstrate the performance of the algorithm compared to other algorithms proposed in the literature. Its main limitation is the lack of statistical testing to validate whether the differences between this method and the others are statistically significant.

---

> ### Author Rebuttal · Authors · 2026-03-30
>
> We sincerely thank Reviewer WQKG for the thoughtful evaluation and for recognizing the originality of our method, the soundness of our theoretical analysis, and the extensiveness of our experimentation. We address the main concern below.
>
> ## Statistical Significance Testing
>
> We thank the reviewer for this valuable suggestion. To further assess the reliability of our results, we repeated the pre-training experiments with **5 random seeds** and report **mean ± standard deviation** together with **two-sided significance tests** against the strongest baseline, **RT**. Due to rebuttal space limitations, we present below a **representative subset of pre-training tasks**; the complete results will be included in the appendix of the revised version.
>
> **Table R1: Statistical significance of RAMBA vs. RT on selected pre-training classification tasks (5 seeds)**
>
> | Dataset | Task | RT | RAMBA | p-value |
> |---|---|---:|---:|---:|
> | rel-amazon | item-churn | 70.1 ± 0.1 | 71.0 ± 0.2 | <0.001 |
> | rel-amazon | user-churn | 63.8 ± 0.3 | 64.5 ± 0.5 | 0.033 |
> | rel-hm | user-churn | 61.9 ± 0.2 | 69.7 ± 0.3 | <0.001 |
> | rel-hm | user-churn | 62.6 ± 0.5 | 69.4 ± 0.1 | <0.001 |
> | rel-stack | user-badge | 79.2 ± 0.2 | 77.9 ± 0.3 | <0.001 |
> | rel-stack | user-engage | 77.3 ± 0.1 | 90.7 ± 0.8 | <0.001 |
> | rel-trial | study-out | 54.3 ± 0.6 | 61.4 ± 0.3 | <0.001 |
>
> **Table R2: Statistical significance of RAMBA vs. RT on selected pre-training regression tasks (5 seeds)**
>
> | Dataset | Task | RT | RAMBA | p-value |
> |---|---|---:|---:|---:|
> | rel-amazon | item-ltv | 32.9 ± 0.4 | 59.5 ± 0.9 | <0.001 |
> | rel-amazon | user-ltv | 33.0 ± 1.2 | 31.2 ± 1.7 | 0.093 |
> | rel-hm | item-sales | 13.0 ± 0.8 | 13.6 ± 1.1 | 0.355 |
> | rel-stack | post-votes | 31.2 ± 1.0 | 34.0 ± 1.0 | 0.002 |
> | rel-trial | site-succ | 2.6 ± 0.4 | 5.6 ± 0.3 | <0.001 |
> | rel-trial | study-adv | -0.0 ± 0.1 | 1.2 ± 0.1 | <0.001 |
>
> These results show that RAMBA's gains are **not attributable to a single favorable seed**. In particular, RAMBA achieves **statistically significant improvements** on several representative tasks, including **rel-amazon/item-churn**, **rel-amazon/user-churn**, both reported **rel-hm/user-churn** settings, **rel-stack/user-engage**, **rel-trial/study-out**, and the regression tasks **rel-amazon/item-ltv**, **rel-stack/post-votes**, **rel-trial/site-succ**, and **rel-trial/study-adv**.
>
> At the same time, the advantage is **not uniform across every task**. For example, the differences on **rel-amazon/user-ltv** and **rel-hm/item-sales** are not statistically significant, and on **rel-stack/user-badge**, **RT performs better**. We believe this heterogeneity is expected in RelBench, where tasks vary substantially in target semantics, schema structure, and the extent to which success depends on global attribute distributions versus more localized relational patterns.
>
> Overall, the significance tests confirm that RAMBA's gains on many key tasks are **robust across random seeds**, rather than artifacts of variance, while also providing a more nuanced picture of where the method helps most. We will include the complete multi-seed results in the appendix of the revised manuscript.

---

> > ### Author Rebuttal · Reviewer_WQKG · 2026-04-06
> >
> > The author have proved the significant differences between results of the RAMBA versus RT.

---

> > > ### Author Response · Authors · 2026-04-08
> > >
> > > We sincerely thank Reviewer WQKG for confirming that our statistical significance analysis has fully resolved the concern. We are grateful for your constructive feedback throughout this process. The addition of rigorous statistical testing has meaningfully strengthened our empirical contribution, and we will ensure these results are included in the final version of the paper. Thank you for your time and careful evaluation of our work.

---

### Official Review · Reviewer_kVMN · 2026-03-13

**Soundness:** 3
**Presentation:** 3
**Significance:** 2
**Originality:** 2
**Overall Recommendation:** 3
**Confidence:** 4

**Summary:**

The paper proposed RAMBA, a model for relational databases that uses selective state-space model. RAMBA incorporates two components, topology-aware linearization and schema dynamic gating. The proposed method show its competitiveness on RelBench, comparing with several baselines.

**Compliance With Llm Reviewing Policy:**

Affirmed.

**Key Questions For Authors:**

-	On Figure 5, how does it look with the classification tasks, instead of the regression task?
-	What is the importance of zero-shot, where the proposed model shows the strength?
-	How does the proposed model perform in terms of training time? (Not just the inference)
-	Are the results of the comparing baselines run or are they taken from the respective papers (as they are compared on the same baseline of RelBench). If they are simply taken from the paper, are the hardware the same across all the methods?
-	What are the fine-tuning settings in terms of the train-size etc.
-	What are some insights (possibly some factors for high performance) for Regressions tasks where RAMBA performs strongly, compared to other baselines?
-	How would RAMBA perform with respect to the extended RelBench?

**Limitations:**

Yes.

**Strengths And Weaknesses:**

-	The paper is easy to follow.
-	The figure captions (Figure 3, 4) can be improved with inclusion of information and description.
-	The results also should show standard deviation, if possible.

---

> ### Author Rebuttal · Authors · 2026-03-30
>
> We thank Reviewer kVMN for the thoughtful review. We address each question below.
>
> **Q1: Figure 5 — Classification for long-context scaling?**
>
> These numbers already appear in Table 8 (Appendix B.1). We summarize:
>
> | Context | RAMBA | RT | Griffin |
> |-|-|-|-|
> | 512 | 72.0 | 69.7 | 63.3 |
> | 1024 | 72.8 | 69.3 | 64.0 |
> | 2048 | 72.0 | 69.3 | 64.5 |
> | 4096 | 72.8 | 69.0 | 63.9 |
>
> RAMBA maintains highest AUROC at every context length. Classification is more saturated (~72–73%), while regression benefits more from longer contexts (+2.0% $R^2$ from 512→4096), as classification often has sufficient signal at shorter contexts.
>
> **Q2: Importance of zero-shot performance?**
>
> Zero-shot generalization is critical for deployment—practitioners cannot retrain for every new schema. RAMBA achieves **72.5% zero-shot AUROC** (vs. 69.3% RT, 64.0% Griffin) and **25.2% zero-shot $R^2$** (vs. 21.9% RT). Key implications: (1) RAMBA's zero-shot already exceeds many baselines' *fine-tuned* performance; (2) stronger initialization leads to faster adaptation—RAMBA reaches 28% $R^2$ within 1k steps, comparable to RT after 10k steps (Figure 6); (3) demonstrates SSMs can serve as relational foundation models. Schema Dynamic Gating enables this by computing gates from schema semantics rather than value distributions, transferring to unseen schemas.
>
> **Q3: Training time comparison?**
>
> | Seq length | RT (tasks/s) | RAMBA (tasks/s) | Ratio |
> |-|-|-|-|
> | 512 | **512** | 208 | 0.41× |
> | 1024 | **253** | 180 | 0.71× |
> | 2048 | 102 | **105** | 1.03× |
> | 4096 | 40 | **56** | 1.40× |
>
> There is a clear crossover: RT trains faster at short contexts due to optimized attention kernels, while RAMBA becomes faster as context grows. Our claim is not universal training speedup, but **better long-context scaling and much stronger inference efficiency**. We will revise the paper to make this distinction explicit.
>
> **Q4: Are baseline results reproduced? Same hardware?**
>
> All results are from **our own reproductions** under a unified setup (same RelBench protocol, same 2×A800 GPUs, BFloat16). Reproduced metrics are within ~1 point of original papers, preserving rankings. We will clarify this explicitly in the revision.
>
> **Q5: Fine-tuning settings?**
>
> Fine-tuning uses **1,024 labeled nodes per task** (few-shot regime), batch size 64, 33k steps, lr $1\times10^{-4}$, AdamW, 2×A800 GPUs. Details are in Section 6.4 and Table 13 (Appendix D.1).
>
> **Q6: Insights on RAMBA's regression performance?**
>
> Four complementary factors: (1) **Column-distribution modeling**: Mamba+column-wise serialization captures global numerical distributions critical for regression. Serialization ablation shows column grouping matters most (95% performance retained even with within-column shuffling). (2) **Schema-aware noise filtering**: Schema Dynamic Gating freezes state for irrelevant columns, crucial since regression is more noise-sensitive (+2.7 $R^2$ over no-gate). (3) **Complementary modules**: Attention-only achieves 72.8 AUROC but only 17.5 $R^2$; adding Mamba yields +7.7 $R^2$, confirming regression needs both structural reasoning and distributional modeling. (4) **Scale-invariant prediction head**: Column statistics normalization (Eq. 12) enables robust cross-table numerical prediction.
>
> **Q7: Performance on extended RelBench?**
>
> We evaluated on **rel-salt** (RelBench v2, 8 ERP classification tasks):
>
> | Task | GraphSAGE | RT | RAMBA |
> |-|-|-|-|
> | item-incoterms | 0.64 | 0.77 | **0.81** |
> | item-plant | 0.99 | 0.99 | **0.99** |
> | item-shippoint | 0.97 | 0.99 | **0.99** |
> | sales-group | 0.20 | 0.23 | **0.26** |
> | sales-incoterms | 0.59 | **0.76** | **0.76** |
> | sales-office | 0.99 | **0.99** | **0.99** |
> | sales-payterms | 0.39 | 0.56 | **0.57** |
> | sales-shipcond | 0.59 | 0.73 | **0.75** |
> | **Average** | 0.67 | 0.75 | **0.77** |
>
> RAMBA consistently generalizes, improving +0.10 MRR over GraphSAGE and +0.02 over RT, with larger gains on harder tasks (item-incoterms +0.04, sales-group +0.03).
>
> **Minor comments**: We revised Figure 3/4 captions to be more descriptive. We also ran 5-seed experiments confirming stability; representative results (mean±std): rel-hm/user-churn AUROC: RT 62.6±0.5 vs RAMBA 69.4±0.1 (p<0.001); rel-amazon/item-ltv $R^2$: RT 32.9±0.4 vs RAMBA 59.5±0.9 (p<0.001). Most improvements are statistically significant. Please see our response to Reviewer WQKG for complete mean±std tables; full results will also appear in the revised appendix.
>
> **Notes on significance**: We emphasize that RAMBA is the **first** work to successfully adapt SSMs to relational databases. The theoretical analysis (Theorem 5.1) formally shows how schema-guided gating achieves favorable SNR scaling as $\text{SNR} = \Omega\left(\frac{(1-\epsilon)n_R}{\delta L}\right)$, independent of context size—a property unique to our approach and critical for real-world databases where relevant signals are sparse among thousands of tokens.

---

> > ### Author Rebuttal · Reviewer_kVMN · 2026-04-04
> >
> > Thanks to the authors for the rebutal. It mostly addressed my concerns, but I have a follow-up question:
> > - What are some concrete examples on the zero-shot? To me, it is difficult to grasp the term zero-shot in some sense, as you already know the number of classes to predict (for classification), for instance. As the paper focuses quite heavily on the zero-shot experiments, it might be useful with more elaborations.

---

> > > ### Author Response · Authors · 2026-04-04
> > >
> > > We thank the reviewer for the follow-up. We provide concrete clarification below.
> > >
> > > **Clarifying "zero-shot" in our context:**
> > > Our zero-shot setup follows the definition established by Relational Transformer [1]: the model is pretrained on six databases and tested on a held-out seventh database **with a completely new schema**, without any weight updates. The model receives only the new database's schema metadata and raw rows through its context window at inference time; it has never seen this database's tables, columns, or value distributions during training.
> > >
> > > **Concrete example:** Consider pretraining on rel-amazon, rel-hm, rel-stack, rel-avito, rel-f1, and rel-trial, then testing on rel-event. The model has never encountered rel-event's tables (e.g., "events", "RSVPs", "users" with event-specific columns like "rsvp_status" or "event_type"). At test time, the model must predict user attendance counts using only schema metadata (column names, table descriptions) and raw cell values from this unseen database. The number of classes or the regression target is defined by the task, but the model must generalize its learned relational reasoning to entirely new table structures and attribute semantics.
> > >
> > > Regarding the reviewer's point about "knowing the number of classes": this is analogous to standard zero-shot classification in NLP/vision, where the output space is defined by the task, but the model has never been trained on data from that domain. The challenge lies in understanding **unseen relational structures and cross-table dependencies**, not the output format. Schema Dynamic Gating is the key enabler: because gates are computed from column-name semantics (e.g., recognizing that "rsvp_status" in the unseen rel-event is semantically related to engagement columns seen during pretraining), the model transfers relational reasoning without memorizing value distributions.
> > >
> > > We acknowledge this terminology deserves clearer exposition. In the revision, we will add a dedicated paragraph with concrete examples (as above) to make the zero-shot setup immediately accessible, beyond the current citation-based description.
> > >
> > > We appreciate the reviewer's thorough engagement throughout this discussion; the questions have genuinely helped us identify areas where the presentation can be improved. We are committed to incorporating all suggested clarifications in the revised manuscript, and we hope these responses have helped address the remaining concerns.
> > >
> > > [1]. Ranjan R, Hudovernik V, Znidar M, et al. Relational Transformer: Toward Zero-Shot Foundation Models for Relational Data. ICLR 2026.

---

### Decision · Program_Chairs · 2026-04-30

**Decision:**

Accept (regular)

**Comment:**

This paper introduces Ramba, a state-space model for relational deep learning that combines topology-aware linearization with schema dynamic gating. The approach proposes a new way to model relational structure while incorporating schema-level information, and reviewers noted its novelty as an early application of state-space models in this domain.

The main concerns raised by reviewers focused on evaluation, robustness, and the role of specific design choices. One issue is the lack of statistical validation in the original submission, with reviewers noting the need to confirm whether observed gains are significant (WQKG). Others questioned aspects of the evaluation setup, including the absence of classification tasks and the practical relevance of the zero-shot setting (kVMN).

A central theme across reviews was sensitivity to schema metadata and sampling design. Multiple reviewers highlighted that the schema dynamic gating mechanism depends heavily on descriptive column names, which may not hold in real-world datasets. As noted, the method can be “highly sensitive to the quality of schema metadata” and may learn incorrect biases when naming is poor (j7xY, 2KWT). Relatedly, concerns were raised about sensitivity to column ordering and whether randomization during training would affect performance (2KWT).

Reviewers also questioned whether the reported gains are tied to specific components such as the width-limited BFS sampling strategy. One reviewer noted that “other baselines when given the same sampling approach might perform similarly” (jtxY), raising the possibility that improvements are not solely due to the proposed model. This concern persisted after rebuttal, with additional observations that performance differences diminish under alternative sampling strategies and that stronger baselines such as HGT were not included in the original comparisons (jtxY).

In the rebuttal, the authors provided substantial additional experiments, including statistical significance testing, ablations on metadata and column order, and extended benchmark results. These additions addressed several key concerns, particularly around robustness to schema design and variability. However, some reviewers maintained reservations about the reliance on width-limited sampling and the completeness of baseline comparisons (jtxY, 2KWT).

Overall, the paper tackles an important problem in a growing new area. The additional experiments significantly strengthen the work and address most of the original concerns. While some limitations remain, the strengths of the approach and the improved empirical validation support a recommendation for acceptance.